# Allosteric regulation of enzymatic catalysis by molecular crowding
Weitong Ren [1], Jiajun Lu[2], Hengyan Huang[2], Jian Zhang[2], Yuxin Chen[3], Zhiqiang Yan [1] ✉, Wenfei Li [1,2,3] ✉ & Wei Wang [2] ✉

Enzymes within cells often function in crowded environments due to the high concentration of macromolecules and the formation of biomolecular condensates. However, most enzyme kinetics insights stem from dilute solution studies. In this study, we examined how molecular crowding impacts enzyme catalysis by constructing a residue-resolved dynamic energy landscape model that explicitly accounts for the crowding agent. By performing molecular simulations for the full enzymatic cycle of a model enzyme adenylate kinase, we revealed a crowding-induced allosteric regulation mechanism of enzymatic catalysis. The crowding agent modifies the conformational distribution of the enzyme, which in turn alters the enzymatic dynamics. Additional simulations of AdK variants with various intrinsic conformational propensities showed that the effect of molecular crowding depends on the rate-limiting step in the enzymatic cycle. Molecular crowding increases enzymatic activity if the rate-limiting step involves the formation of the catalytically competent enzyme-substrate complex and decreases the activity if the rate-limiting step involves the product release. The results revealed in this work not only shed insights into the general biophysical principle of protein dynamics under a crowded cellular environment but also provide a computational framework to understand the diversity of existing experimental observations of the molecular crowding effects on enzyme catalysis.

Cellular environment is typically highly crowded, with ~30% of the volume occupied by a diverse array of molecules[1,2]. These include not only small metabolites but also large biomacromolecules such as proteins, nucleic acids, glycans, and their complexes, spanning a wide range of molecular sizes and shapes. Such a crowded and heterogeneous cellular milieu is a fundamental physical feature of living cells and is known to exert profound effects on biomolecular dynamics and function by modulating conformational motions, intermolecular interactions, and other molecular processes[3–6]. Beyond this baseline level of heterogeneous crowding in the cytosol, cells actively regulate and organize crowding effects through higher-order spatial and functional compartmentalization via the formation of biomolecular condensates. These condensates represent dynamically organized, highly concentrated assemblies that impose well-defined structural organization, selectively control molecular composition and mobility, and thereby fine-tune biochemical reaction rates, signaling pathways, and enzymatic activities[7–10].

Over the past few decades, extensive research has been conducted to unravel the molecular mechanisms governing biomolecular dynamics under crowded cellular environments. A significant focus has been on understanding how crowding conditions influence protein conformational dynamics, including protein stability, folding, and allostery[3,5,11,12]. It is widely acknowledged that a primary effect of crowded environments is the stabilization of compact biomolecular conformations via entropic effects driven by volume exclusion[3,5,11]. Other studies suggest that crowding can also regulate biomolecular dynamics through additional factors, such as soft interactions between crowding agents and solutes, agent shape, and the composition of the crowded environment[12]. These studies demonstrated the complexity in the biophysical contributions to the effects of molecular crowding on biomolecular dynamics.

The impact of molecular crowding on enzyme catalysis is even more intricate and less understood. Enzymatic activity is tightly coupled to multiple dynamic steps, including enzyme conformational changes, substrate binding, product release, and chemical reactions, making it particularly difficult to elucidate the physical principles governing catalysis in crowded environments. Over the past two decades, experimental studies have investigated enzyme activity under crowded conditions, revealing

[1]Wenzhou Key Laboratory of Biophysics, Wenzhou Institute, University of Chinese Academy of Sciences, Wenzhou, PR China. [2]School of Physics, National Laboratory of Solid State Microstructure, Nanjing University, Nanjing, PR China. [3]Department of Cardiology, Jiangsu Key Laboratory for Cardiovascular Information and Health Engineering Medicine, Nanjing Drum Tower Hospital, Medical School, Nanjing University, Nanjing, PR China. ✉e-mail: zqyan@ucas.ac.cn; wfli@nju.edu.cn; wangwei@nju.edu.cn

diverse and sometimes even opposite outcomes. For some enzymes, such as phosphoglycerate kinase (PGK)[13], various ribozymes[14–17], DNA polymerase[18], molecular crowding has been shown to enhance activity, which can be related to the stabilization of catalytically competent enzyme-substrate complexes[13–25], reduced internal friction[26], and alleviation of product inhibition[27]. Conversely, molecular crowding has been observed to decrease the activity of other enzymes, such as α-chymotrypsin, horseradish peroxidase, and lactate dehydrogenase[28–34], which can be attributed to the lowered accessibility of active sites[32], the activation-diffusion control mechanism[30], the hindered chemical reactions and product release[31,34], the stabilization of inert conformations[33], and the enhanced product inhibition[28]. Non-monotonic dependence of catalytic activity on crowding volume fraction was also found in multi-cooper oxidase due to changed protein internal dynamics and increased solution viscosity[35].

Complementary theoretical and computational studies further underscore the complexity of crowding effects on enzymatic catalysis. Pioneering work by Minton demonstrated that excluded-volume effects generally reduce enzyme-catalyzed reaction rates while promoting biomolecular association through confinement[36,37]. McCammon and Zhou later showed that crowding can slow the in vivo activity of HIV protease by suppressing flap dynamics and reducing enzyme-ligand encounter rates[38,39]. In contrast, studies by Kondrat and co-workers revealed that crowding can accelerate channeled cascade reactions and modulate the binding behavior of divalent biomolecules, as predicted by scaled-particle theory and Brownian dynamics simulations[40,41]. Despite these advances, most theoretical approaches focus on individual steps, such as conformational fluctuations, ligand binding, or molecular diffusion, rather than addressing the fully coupled, multistep nature of enzymatic catalysis. Taken together, the diverse and sometimes opposing effects of molecular crowding on enzyme activity underscore the difficulty of achieving a unified mechanistic understanding

and highlight the need for integrative approaches that explicitly account for the interplay between enzyme dynamics, reaction kinetics, and the crowded cellular environment.

One extensively studied model enzyme is adenylate kinase (AdK), which catalyzes the reversible transfer of a phosphate group from ATP to AMP, producing two ADPs. AdK consists of three domains, including CORE, LID, and NMP domains[42–51] (Fig. 1A). The substrate ATP and AMP binds at the binding pockets located at the LID-CORE interface and NMP-CORE interface, respectively. In the apo state (without substrate and product), the enzyme preexists as an equilibrium between open and closed conformations (Fig. 1A, B), with the open state being thermodynamically dominant. Substrate binding to their respective sites shifts this pre-existing equilibrium by stabilizing the closed conformation through favorable interactions that reshape the free energy landscape, forming a catalytically competent state for phosphate transfer. After the phosphoryl transfer reaction occurs in the closed conformation, the enzyme reopens to release the products (see Fig. 1C). Each catalytic turnover is therefore associated with one complete cycle of domain opening and closing. AdK's activity primarily relies on coupled substrate binding, protein conformational changes, chemical reactions, and product release. This enzymatic activity is particularly sensitive to its conformational equilibrium, which can be modulated by mutations[52,53], temperature adaptation[54,55], addition of TMAO[52] or urea[50,56], and the introduction of a disulfide bond between the LID and NMP domains[57]. Experimental findings demonstrate that shifting the conformational population towards the closed conformation by mutation or adding cosolvent decreases catalytic activity[52,54,57], while a shift towards the open conformation increases catalytic activity[52,53,55,56]. Single-molecule and NMR experiments revealed that wide range of timescales involved in the conformational motions of the enzyme[44,47,58]. Our previous computational studies revealed that this enzyme exploits steric frustration to

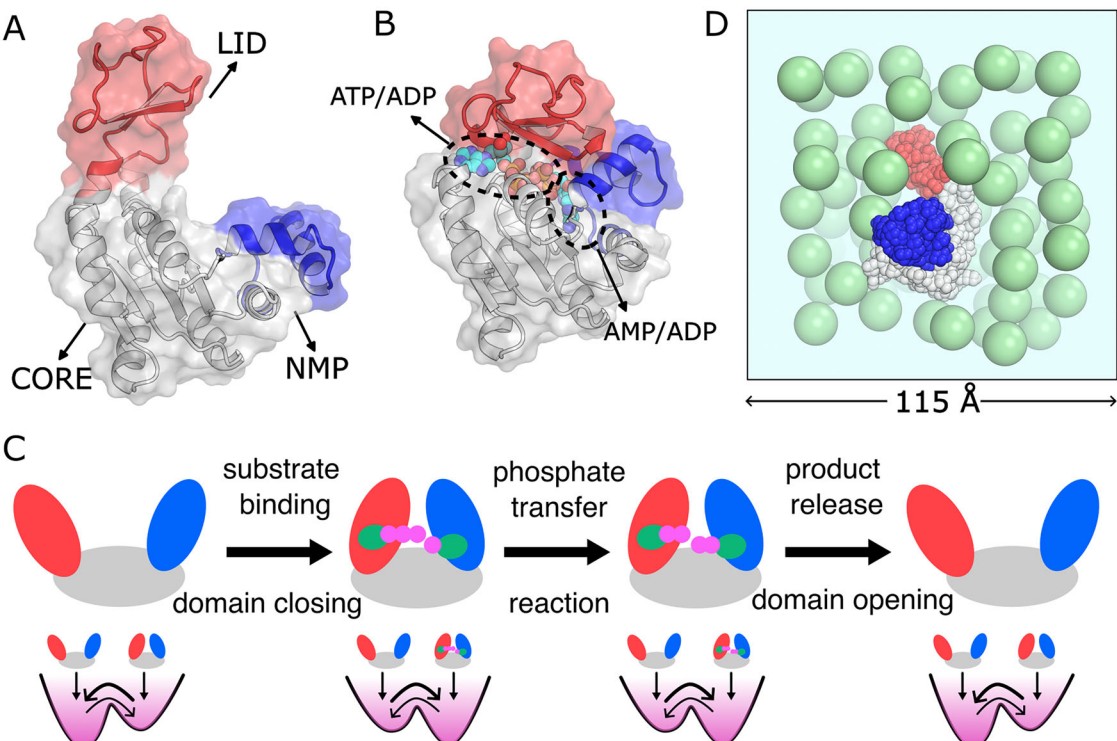

**Fig. 1 | Structural domains, catalytic cycle, and crowded simulation setup of AdK. A, B** The open (pdb code: 1AKE[78]) and closed (AP5-bound, pdb code: 4AKE[77]) conformations of wild-type AdK. The CORE, LID, and NMP domains are marked in gray, red, and blue, respectively. AP5, an analog of the substrate ATP and AMP, is shown explicitly in a sphere representation. After the chemical reaction, the ATP and AMP sites are occupied by ADP. The binding sites for ATP and AMP are indicated by dashed epsilon circles. **C** Schematic representation of the AdK catalytic cycle and the corresponding alterations in the underlying free-energy landscape. **D** An illustration of the crowded system with a volume fraction($\phi$) of 0.3. The crowder particles are represented by spherical beads in green. The simulation graph was prepared by PyMOL[80].

facilitate the rate-limiting product release step and that repeated conformational transitions are crucial to correcting the nonnative substrate binding state in the enzymatic cycle[51,59,60]. These well-characterized properties of AdK make it a perfect model enzyme for investigating the general molecular mechanism of enzyme catalysis under a crowded environment.

In this study, we investigated the molecular mechanism of the crowding effect on enzyme catalysis by using AdK as a model enzyme with residue-level coarse-grained (CG) molecular dynamics (MD) simulations. MD simulations have been widely utilized to investigate the conformational dynamics involved in enzyme catalysis due to their inherent advantage in temporal and spatial resolution. However, conventional all-atom MD simulations often encounter computational bottlenecks when applied to investigate the full cycle of enzyme catalysis, in which the individual steps have tight interplay and involve a wide range of timescales. In our previous work, we developed a residue-resolved dynamic energy landscape model to tackle the multiscale difficulty in enzyme catalysis[51]. Building on this model, here we further construct a computational framework with explicit consideration of the crowding agent and use it to investigate the influence of crowding on complex enzymatic reactions, which have been largely overlooked in current studies. Molecular simulations for AdK revealed a crowding-induced allosteric regulation mechanism of enzymatic catalysis, whereby molecular crowding alters the enzymatic activity by modulating the conformational motions. While molecular crowding typically stabilizes the compact closed state through volume exclusion, the effect of molecular crowding on the catalytic cycle can vary, depending upon the intrinsic conformational propensity of the enzyme and the corresponding rate-limiting step of the enzymatic cycle. The same crowding agent may have the opposite effect on the enzyme models with a different intrinsic conformational propensity, which sheds new insights into the molecular mechanism of the diverse effects of molecular crowding on enzyme activity typically observed in previous biochemical experiments.

## Results

### Molecular modeling of the enzymatic cycle under crowding

In our previous studies, we established a dynamic energy landscape framework at residue-level resolution to model the full enzymatic cycle, effectively capturing the coupling among enzyme conformational dynamics, ligand binding, and chemical reactions at the residue level[51]. In this approach, each amino acid residue is represented by a single particle positioned at its $C_\alpha$ atom. This simplified representation significantly reduces computational cost while retaining the essential topology of the protein, making it feasible to simulate large-scale, functionally relevant conformational transitions. The framework is based on the conformational selection (or population shift) mechanism, which posits that the enzyme intrinsically samples both open and closed conformational states even in the absence of ligands[61,62]. Ligand binding selectively stabilizes pre-existing closed states, thereby shifting the conformational equilibrium toward the catalytically competent ensemble. In addition, the rate-limiting step in the AdK catalytic cycle is ligand unbinding coupled with conformational transition[63]. The energy function comprises two components that jointly encode how ligand occupancy modulates the enzyme's conformational landscape:

$$V(\boldsymbol{r}, \{ls_1, ls_2, \ldots\}) = V_{apo}(\boldsymbol{r}, \Delta V) + \sum_i V_{bind}(\boldsymbol{r}, ls_i) \quad (1)$$

Here, $V_{apo}(\boldsymbol{r}, \Delta V)$ approximates the intrinsic energy landscape of the enzyme in the absence of ligands (apo state), formulated as a structure-based potential with a multi-basin topology that reflects the coexistence of distinct conformational states—such as open and closed forms[51,64–66]. The parameter $\Delta V$ controls the relative stability between these states and can be calibrated to match experimental estimates of their equilibrium populations[51,67]. The ligand-dependent energy term $V_{bind}(\boldsymbol{r}, ls_i)$ accounts for the energetic effect of ligand binding at the $i$-th binding site $ls_i$ in an implicit manner: rather than modeling ligands as explicit particles, their presence is represented by the formation of specific residue-residue attractive contacts in the binding site of

the holo (ligand-bound) structure but absent in the apo state. AdK possesses two distinct binding pockets: one specific for the AMP or ADP and the other for the ATP or ADP. In functional states, these sites are typically occupied either by AMP and ATP (substrate) or by two ADP molecules (product). Each pocket can independently be occupied by a substrate, a product, or remain empty, resulting in up to nine possible ligand occupancy combinations. Importantly, the ligand-dependent energy term $V_{bind}(\boldsymbol{r}, ls_i)$ differs depending on whether a substrate or a product is bound, as each induces a distinct set of stabilizing contacts in the binding pocket. Consequently, each binding state corresponds to a unique energy landscape that reflects the specific interactions associated with the bound species. Changes in ligand state—through binding, unbinding, or chemical reaction—dynamically reshape the underlying energy landscape, thereby driving the progression of the catalytic cycle. Ligand binding and unbinding are modeled via a Metropolis Monte Carlo scheme with the rates depending on ligand concentrations, binding energies, and solvent accessible surface area(SASA) of the binding pockets[51]. Warshel et al. showed that protein conformational motions facilitate chemical step mainly by preorganizing the active sites to the catalytically competent state[68]. We therefore implement the catalytic reaction step through a kinetic Monte Carlo scheme with the inputs of the experimentally determined forward and reverse reaction rates[63]. The chemical reaction is possible only when the enzyme samples a closed conformation with two substrates bound and the residues of the active sites are optimally positioned via conformational motions. The above dynamic energy landscape framework enables efficient simulations of the tight interplay between the individual chemical and physical steps in the full enzymatic cycle without going into the microscopic details of the chemical event, which otherwise requires quantum mechanics treatment.

We explicitly model the molecular crowding by adding crowding agents in the simulation box. The crowding agents are represented by inert spherical particles. Following previous works[69,70], the crowder-crowder and crowder-enzyme interactions are described by an excluded volume effect term. The crowding is systematically controlled by adjusting the occupied volume fraction $\phi$ in the simulation box. This minimalist approach captures essential excluded volume effect while maintaining computational efficiency and easy to implement, allowing quantitative investigation of how the crowding environment impacts enzyme dynamics. We also investigated the case with a soft crowding agent by introducing non-specific attractive interactions. More details of the model construction and molecular simulations of enzyme dynamics under crowding are described in Methods and Supplementary Materials.

### Effect of molecular crowding on enzyme activity of wild type AdK

Firstly, we investigated the impact of the presence of inert crowders (with excluded volume interactions) on the activity of the wild-type (WT) AdK. We conducted 20 independent simulations for the WT AdK under different crowded conditions and substrate concentrations, with crowding volume fractions of 0, 0.1, 0.2, 0.3, and 0.4. By analyzing the simulation trajectories, we can derive the turnover rates. As depicted in Fig. 2A, the predicted turnover rate $v$ of AdK, plotted against ATP concentration [ATP], aligns well with the Michaelis–Menten equation $v = \frac{k_{cat}[\text{ATP}]}{K_M+[\text{ATP}]}$, with $k_{cat}$ and $K_M$ being the maximum turnover rate and Michaelis constant, respectively. Elevating the crowding volume fraction decreases the AdK turnover rate. By fitting the turnover rate against ATP concentration with the Michaelis–Menten equation, we can extract the key enzymatic parameters $k_{cat}$ and $K_M$. As expected, the crowded environment resulted in a reduction in the $k_{cat}$, with a more crowded environment leading to a more pronounced slowdown of catalysis (Fig. 2B), signifying the dampening of enzyme activity in the presence of inert crowders. In contrast, $K_M$ shows a slight decrease under low and moderate crowding, whereas under high crowding conditions $K_M$ begins to increase. $K_M$ is often associated with substrate affinity in simple Michaelis–Menten kinetics, and the small decrease observed at moderate crowding may reflect stabilization of the closed conformation, which generally exhibits higher substrate affinity[52,57]. At high crowding, excessive stabilization of closed states can favor the accumulation of non-

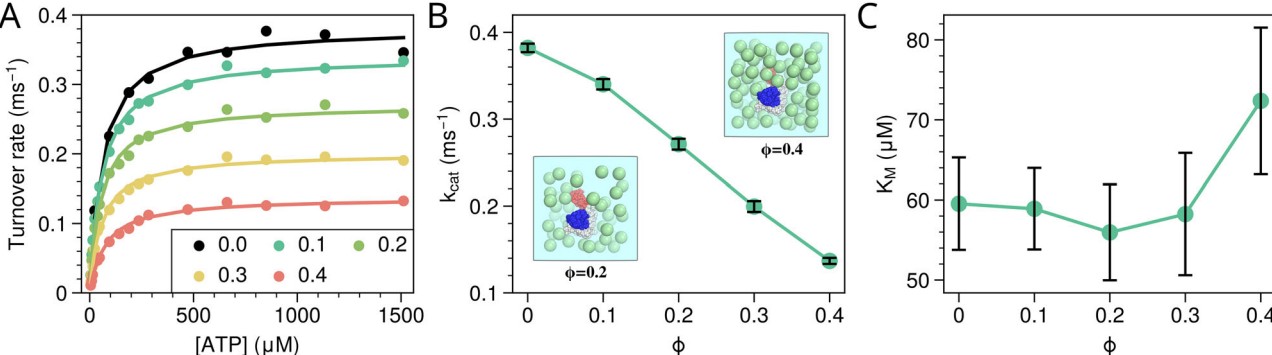

**Fig. 2 | Effect of crowding agent on the catalytic activity of wild-type AdK.**
**A** Turnover rates as a function of ATP concentrations at a fixed AMP concentration of 300 μM with different crowding volume fractions. Solid lines represent the fits to the Michaelis–Menten equation. The kinetic parameters $k_{cat}$ (**B**) and $K_M$ (**C**) derived from the Michaelis–Menten fits as a function of crowding volume fraction. Error bars are determined using bootstrapping.

productive ligand-bound closed conformations, thereby reducing the population of catalytically competent states, as illustrated in Supplementary Fig. S1. However, the fitted $K_M$ values show substantial uncertainty, and the observed changes are small; therefore, these trends should be interpreted with caution and not overemphasized. It is worth noting that the large uncertainty in $K_M$ is primarily attributable to fitting errors. According to the Michaelis–Menten equation, $K_M$ is predominantly determined by data points at low substrate concentrations. In our study, however, measurements at such low concentrations were limited. Consequently, the bootstrap analysis used to estimate uncertainty yields relatively large errors, leading to the observed variability in $K_M$.

### Effect of molecular crowding on enzyme activity for AdK variants with different intrinsic conformational propensity

Next, we investigate the effect of crowding on the catalytic kinetics of AdK variants with distinct conformational propensity. As discussed above, AdK dominantly adopts two conformations, i.e., closed conformation and open conformation. The intrinsic conformational propensity can be quantified by the relative population of the closed conformation ($P_{closed}$) at the apo state in the dilute solution. In the above discussion on the WT AdK, the energy gap $\Delta V_i$ was optimized according to experimentally measured $P_{closed}$ (=0.3)[44]. By tuning $\Delta V_i$ values in the dynamic energy landscape model, we can construct AdK variants with different intrinsic conformational propensity (Supplementary Table S1). AdK variants with larger (smaller) $P_{closed}$ values correspond to closed-biased (open-biased) enzyme models. It is important to note that this approach creates theoretical models that mimic the energetic effects of mutations without involving actual amino acid changes, and do not directly correspond to real AdK mutants. Figure 3A compares the turnover rates for the AdK variants with different conformational propensity in dilute ($\phi = 0$) and crowded environments ($\phi = 0.3$). Interestingly, we observed that the impact of the crowding is closely related to the conformational propensity of the AdK models. When $P_{closed}$ is very low (i.e., AdK predominantly adopts the open conformation), the enzyme has a low turnover rate in dilute conditions. The introduction of a crowded environment with $\phi = 0.3$ significantly increases the turnover rate. For AdK variants with a relatively larger $P_{closed}$ (> 0.01), the catalytic activity is markedly suppressed in crowded conditions.

By fitting the catalytic kinetics with Michaelis–Menten equation, we extracted $k_{cat}$ and $K_M$ for the AdK variants with different $P_{closed}$ at dilute and crowded conditions (Fig. 3B, C). In dilute solution, the $k_{cat}$ value exhibits a biphasic trend. It increases when $P_{closed} < 0.05$, reaches a maximum at $P_{closed} \sim 0.05$, and decreases when $P_{closed} > 0.05$. Upon introducing crowders, the enzymatic activity monotonically diminishes with $P_{closed}$. Consequently, adding crowding agents has an opposite effect on $k_{cat}$ for the AdK variants with different conformational propensities. For AdK with open-biased conformations, adding crowding agents increases $k_{cat}$, suggesting crowding activation of catalytic kinetics. On the contrary, for AdK variants with closed-biased conformation, adding crowding agents decreases $k_{cat}$,

corresponding to crowding inhibition of catalytic kinetics. The $K_M$ value exhibits a monotonic decreasing trend with $P_{closed}$ both in the dilute ($\phi = 0$) and crowded environments ($\phi = 0.3$). For AdK variants with the open-biased conformations, adding crowder significantly decreases $K_M$. This crowding effect on $K_M$ becomes negligible for the AdK variants with the open-biased conformations ($P_{closed} > 0.3$).

To further investigate the impact of the inert crowder on enzyme activity, we systematically computed the turnover rates of various AdK variants under diverse crowded environments at a fixed ATP concentration of 300 μM. The obtained turnover rates are plotted against $\phi$ and $P_{closed}$ (Fig. 4). In the absence of crowder particles in the solution, the enzyme exhibits high activity across a broad range of $P_{closed}$. Upon the gradual addition of crowd particles to the solution, the region of high activity shifts toward lower $P_{closed}$ values and progressively narrows. Furthermore, the maximum catalytic activity decreases monotonically with increasing $\phi$.

The effect of crowding depends strongly on the intrinsic conformational bias of the enzyme. For variants with $P_{closed} > 0.05$, increasing crowding leads to a monotonic reduction in enzymatic activity, with physiologically relevant crowding ($\phi \approx 0.3$) causing an approximately 50% decrease. This result is consistent with the predictions of Skora et al., who reported that at physiologically relevant occupied volume fractions of 20–30%, enzymatic activity can be reduced by 40–50%[40]. In contrast, for extreme open-biased variants ($P_{closed} \approx 0.003$), the introduction of crowding results in a monotonic enhancement of activity. For intermediate variants, crowding produces a biphasic response, in which low crowding enhances activity, whereas high crowding suppresses it. As discussed in the following section, this non-monotonic behavior arises because substrate binding must precede catalysis. Moderate crowding stabilizes the closed conformation, shifts the conformational equilibrium toward this catalytically competent state. Additionally, it speeds up the domain closing motion itself. These effects enhance the probability of forming the active, substrate-bound complex, thereby accelerating the overall rate of reaching the catalytically competent state (i.e., the apparent rate of substrate binding). In contrast, excessive crowding over-stabilizes the closed conformation. This causes the enzyme to close too quickly often before the substrate binding, and impedes product release. Consequently, these factors hinder effective substrate binding and disrupt the completion of the catalytic cycle.

Together, these results demonstrate again that the crowding effect on catalytic activity is sensitive to the intrinsic conformational propensity of enzymes. The same crowding agents may impose opposite effects on the enzymatic kinetics for the enzymes with different intrinsic conformational propensity.

### Effect of molecular crowding on the individual steps in the enzymatic cycle
The catalytic cycle of AdK consists of multiple physical and chemical steps, including substrate binding, conformational motions, chemical reaction,

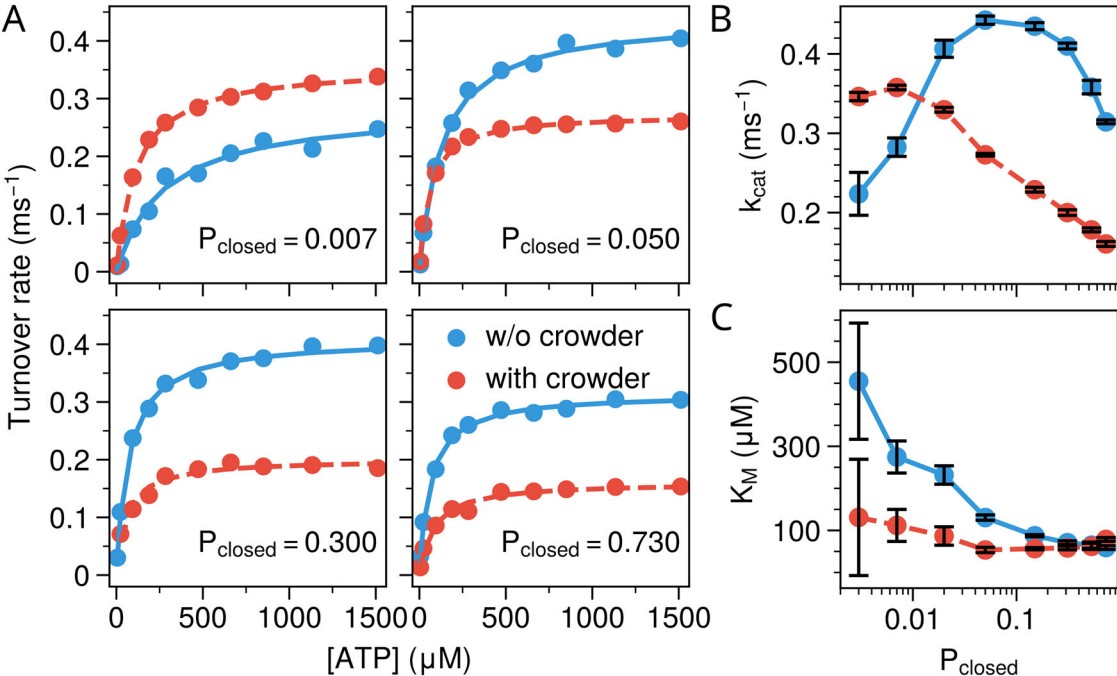

**Fig. 3 | Impact of molecular crowding($\phi = 0.3$) on the catalysis of various AdK variants with different conformational propensities ($P_{closed}$). A** Turnover rates as a function of ATP concentration for AdK variant models with varying conformational propensity ($P_{closed}$) in dilute solution (blue) and crowded environments (red). Solid and dashed lines represent Michaelis–Menten fits for the dilute and crowded conditions, respectively. Enzyme kinetic parameters $k_{cat}$ (**B**) and $K_M$ (**C**) as functions of $P_{closed}$ under dilute (blue) and crowded (red) conditions. Error bars represent fitting uncertainties calculated using bootstrapping.

**Fig. 4 | Turnover rates of AdK variant models with different conformational propensity ($P_{closed}$) at different crowding volume fractions ($\phi$).** Turnover rates were calculated at a substrate concentration of [ATP] = [AMP] = 300 $\mu$M.

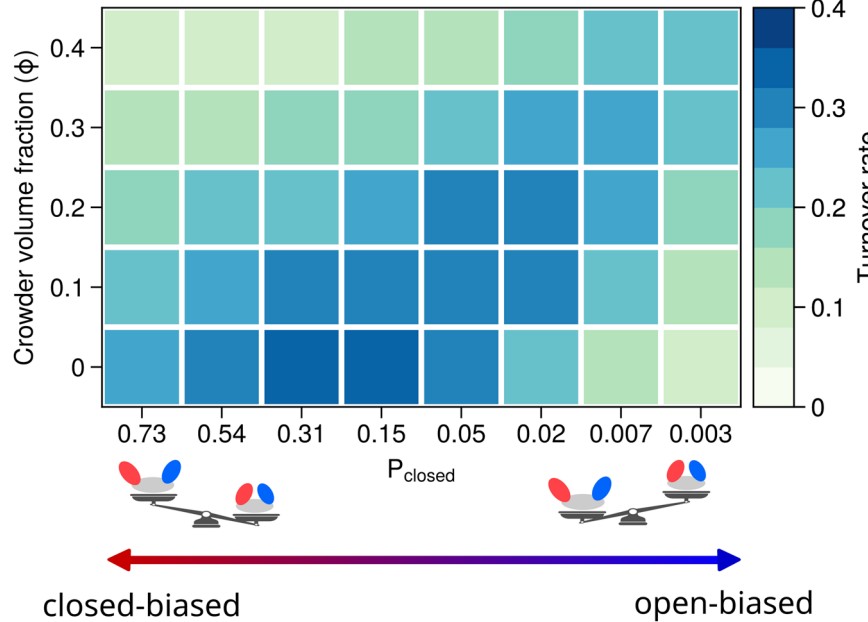

and product release. The chemical reaction itself often occurs much faster once the enzyme adopts a catalytically competent state. To more deeply elucidate the molecular mechanism underlying the crowding-modulated enzymatic catalysis, we separately investigate the kinetics of the conformational motions, the substrate binding, and the product release under different crowding conditions.

We first investigate the effect of molecular crowding on the enzyme conformational dynamics. To examine this, we utilized two collective variables, i.e., the center-of-mass(COM) distance between the LID and CORE domains($R_{LID-CORE}$) and the COM distance between the NMP and CORE domains($R_{NMP-CORE}$), to track the movements of the LID and NMP domains, respectively. Figure 5A compares the two-dimensional free energy landscape of the wild-type AdK along these two collective variables in dilute and crowded environments. In both conditions, besides the two most stable states (open and closed), there are two metastable intermediate states, in which only one of the two domains is open. The introduction of crowder particles at $\phi = 0.3$ alters the free energy landscape by significantly increasing the relative stability of the closed conformation. More systematic analysis across various AdK variants at different crowding volume fractions showed that the presence of crowding consistently increases the population of the

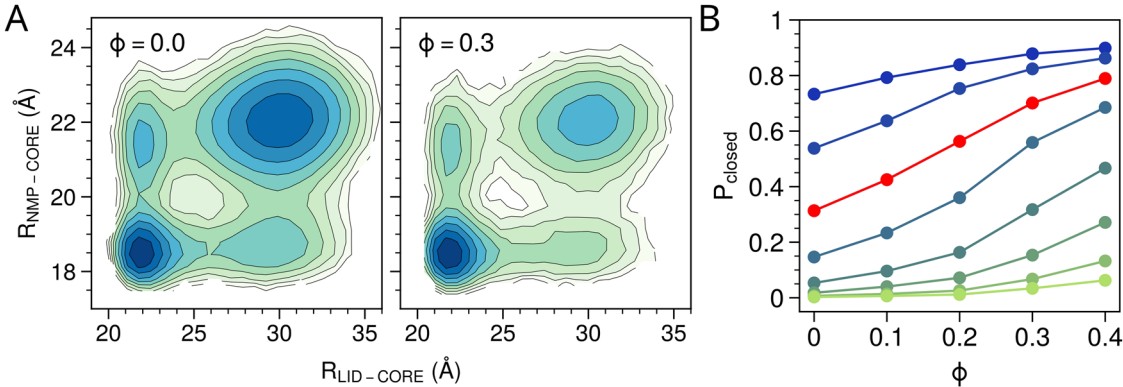

**Fig. 5 | Allosteric regulation of enzyme conformation of AdK models by molecular crowding. A** Two-dimensional free energy landscape of wild-type AdK along the LID-CORE distance($R_{LID-CORE}$) and NMP-CORE distance($R_{NMP-CORE}$) in the dilute solution (left) and the crowded environment with $\phi = 0.3$ (right). The concentrations of ATP and AMP are fixed at 300 μM. **B** Observed probability of the closed conformation $P_{closed}$ of various AdK variant models at different crowded conditions. Different lines represent the results for the AdK variant models with different conformational propensity. The WT AdK result is shown in red.

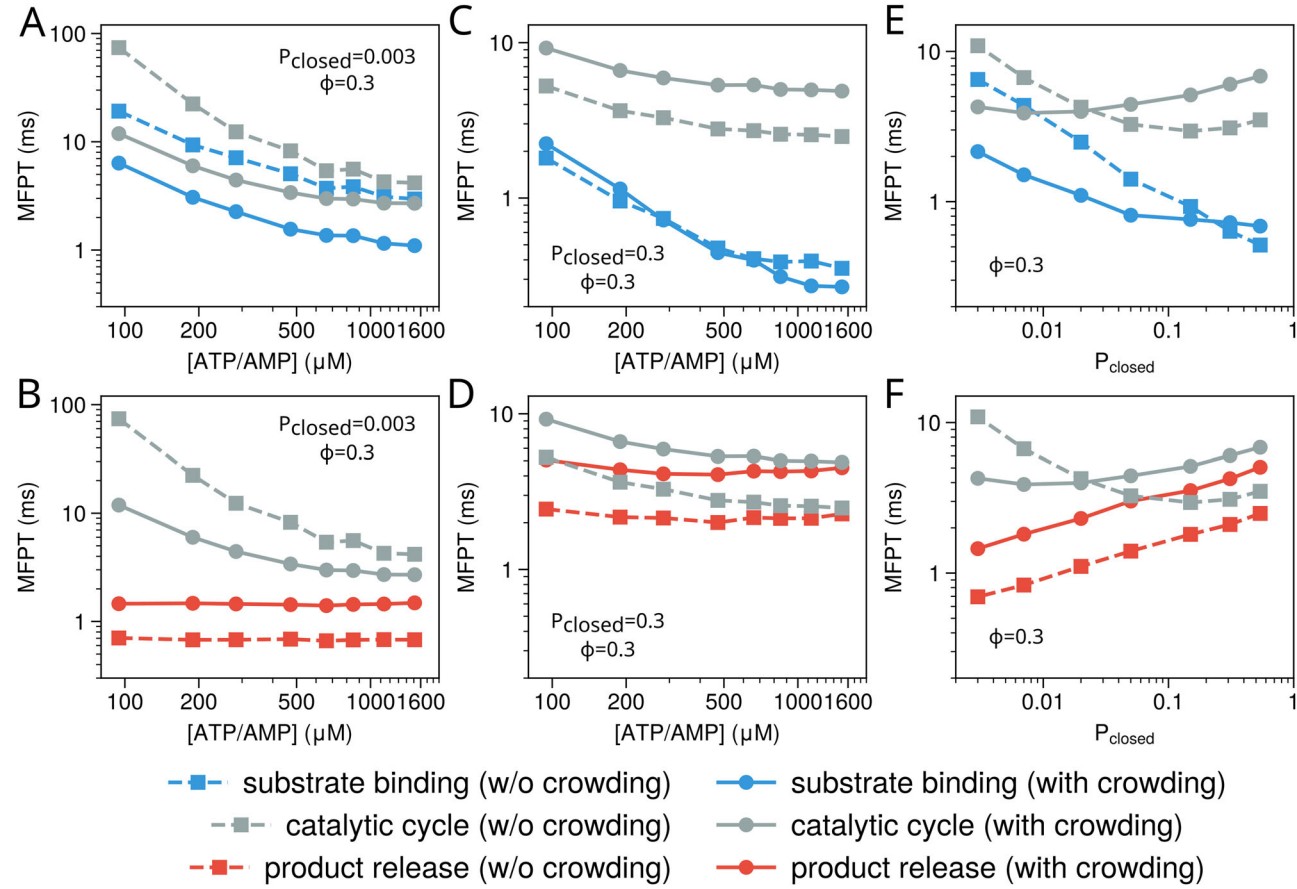

**Fig. 6 | The effect of molecular crowding on the separated catalytic steps.** The mean first passage time (MFPT) for the enzyme to reach the catalytically competent state starting from the open, substrate-free state (**A**), and MFPT for product release from the closed, product-bound conformation (**B**) as functions of substrate (ATP) concentrations for the AdK variant model with $P_{closed} = 0.003$. For comparison, the turnover times are also plotted for reference (gray). The results at dilute condition and with molecular crowding are shown in solid lines and dash lines, respectively. **C**, **D** Same as (**A**, **B**) but for the WT AdK model ($P_{closed} = 0.3$). MFPT for the enzyme to access the catalytically competent state (**E**) and for product release (**F**) as a function of $P_{closed}$ at [ATP] = [AMP] = 300 μM.

closed state, with higher volume fractions leading to greater enhancement (Fig. 5B).

We then performed simulations for the WT AdK($P_{closed} = 0.3$) and the AdK variant with an open-biased conformational propensity ($P_{closed} = 0.003$) at different substrate concentrations, both with and without molecular crowding. As expected, the mean first passage time (MFPT) for the enzyme to access the closed state with two substrate correctly bound (hereafter referred to as substrate binding step) decreases as substrate concentration increases, while the MFPT for the product release is insensitive to substrate concentrations(see Fig. 6C, D). For the open-biased AdK model, the crowding environment accelerates the substrate binding step but slows down the product release step. The acceleration of the substrate binding step primarily arises from the crowding enhanced open-to-closed conformational transition, as

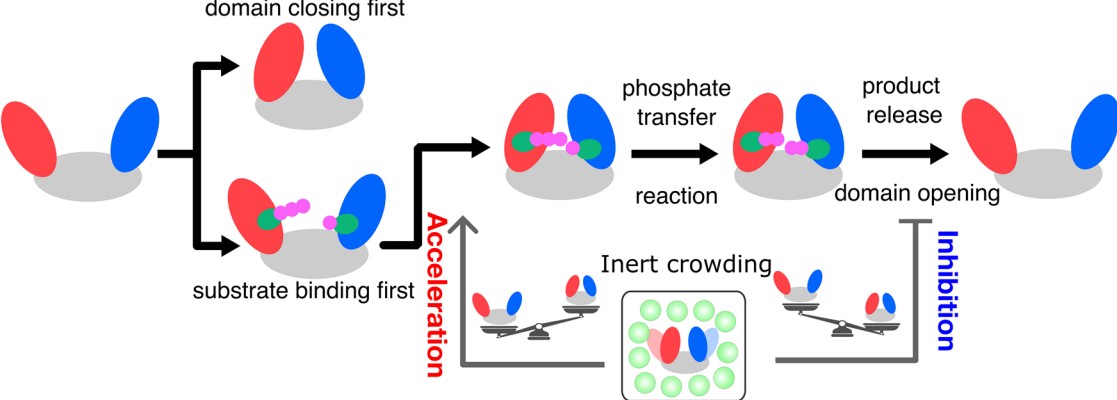

**Fig. 7 |** Schematic diagram illustrating the molecular mechanism of molecular crowding on the catalytic activity for the enzyme models with different conformational propensity.

the substrate binding rate itself is independent of crowding in our model (Fig. 6). Similarly, adding crowder enhances the stability of the closed conformation, which leads to the slowdown of the product release. As the substrate binding step is much slower than the product release step and therefore corresponds to the rate-limiting step in the catalytic cycle, adding crowding agent increases the overall turnover rate at a wide range of substrate concentrations (Fig. 6A). In contrast, for the WT AdK model, the product release step corresponds to the rate-limiting step at moderate and high substrate concentrations. The presence of molecular crowding($\phi = 0.3$) slows down the product release step, and therefore decreases the overall turnover rate.

The above results illustrate distinct crowding effects for the WT AdK model and open-biased AdK model. More systematic analysis showed that adding crowding tends to accelerate the substrate binding step for a wide range of open-biased AdK models, but the effect becomes negligible when $P_{closed}$ becomes large (>0.1) (Fig. 6E, F). In comparison, introducing crowding consistently slows down the product release process for all AdK variants. Because the rate-limiting step switches from substrate binding to product release with the increasing of $P_{closed}$ at the substrate concentration of 300 μM, adding crowding leads to opposite effects on the overall catalytic kinetics for the open-biased AdK model and closed-biased AdK model.

**Impact of crowder size and soft crowder interactions on enzyme dynamics**

As noted earlier, cellular crowding is a highly heterogeneous environment composed of biomolecules of diverse sizes and physicochemical properties. Previous studies have shown that both the size and the chemical nature of crowders play important roles in modulating protein behavior[12]. To further clarify these contributions, we performed additional simulations of WT AdK under crowded conditions at $\phi = 0.3$ using larger crowder particles ($r = 12$ Å). Interestingly, these larger crowders produced a weaker excluded-volume effect (Supplementary Fig. S2), consistent with the expectation that larger crowders suppress enzymatic activity less strongly than smaller ones.

We also investigated soft crowders that include attractive crowder-enzyme interactions. These interactions influence the catalytic dynamics of both WT and closed-biased AdK in a strength-dependent manner. For WT AdK, soft crowding generally slows catalysis, with stronger attraction and higher volume fractions producing greater inhibition (Supplementary Fig. S3). Weak soft crowding ($\epsilon_{attr} = 0.2$) slightly enhances WT activity compared to inert conditions, whereas strong soft crowding ($\epsilon_{attr} = 0.3$) markedly suppresses activity and nearly abolishes it at high volume fractions.

The closed-biased variant exhibits a more complex response (Supplementary Fig. S4): moderate soft attractions enhance its activity relative to inert crowding, though not to dilute-solution levels. Under strong soft

crowding, however, its activity exceeds even that in dilute conditions. These results suggest that soft crowding can fine-tune enzymatic activity in a manner that depends on both interaction strength and the enzyme's conformational preference.

We emphasize that these additional simulations are intended to illustrate the rich and nontrivial nature of crowding effects on enzyme dynamics, rather than to provide a comprehensive characterization of all possible crowding regimes.

**Discussion and conclusion**

Despite numerous experimental and theoretical efforts over the last few decades, how molecular crowding modulates enzymatic dynamics remains a poorly understood question. In many cases, different and sometimes even opposite effects were observed for the same enzyme under different crowding agents or different enzymes in the same crowded environment. The residue-resolved dynamic energy landscape model facilitates single-molecule-level molecular simulations of the full catalytic cycle involving the tight interplay among all the physical and chemical steps under a crowding environment, which is otherwise unfeasible using all-atom MD simulations and quantum mechanical approaches. This full-cycle molecular simulation enables the elucidation of the effect of molecular crowding on the individual steps of the catalytic cycle and the underlying biophysical mechanism. In general, macromolecular crowding with inert crowders tends to modify the free energy landscape by stabilizing the compact conformation attributed to the entropic effect. Following the similar line, we observed that the impact of crowding on enzyme activity is closely linked to the enzyme conformational propensity and the rate-limiting step in the catalytic cycle. Distinct effects of crowding are observed for the enzymes with different conformational propensity and rate-limiting step (Fig. 7). When the closed state is highly unstable, and substrate binding corresponds to the rate-limiting step, the presence of an inert crowder tends to enhance enzymatic activity by expediting substrate association and conformational closing, which is the prerequisite for the enzyme to sample the catalytically competent state. Conversely, if the closed state is relatively stable and product release from the closed state after catalytic reaction becomes difficult and therefore determines the overall catalytic rate, crowding is more likely to suppress enzyme activity by excessively stabilizing the closed conformation, further hindering product dissociation. These findings demonstrate an allosteric regulation mechanism of enzymatic catalysis under molecular crowding, and the results suggest that it is possible to estimate enzyme activity in a crowded environment based on its conformational dynamics and enzyme activity in a dilute solution. Furthermore, these results provide guidance for the modification of existing enzymes or the design of new enzymes that display higher activity within a crowded environment.

In experiments, polymers such as ficoll and dextran are frequently employed as crowding agents to simulate a crowded environment, which

modulates the enzymatic dynamics primarily by the excluded volume effect. However, in the cellular context or within biomolecular condensates formed by biomolecular liquid-liquid phase separation, the crowding environment contains densely packed proteins or nucleic acids. Non-specific attractive interactions, including dispersion forces, electrostatic forces, and hydrogen bonding, may take place between enzymes and these surrounding biomolecules. Due to the intricate nature of these interactions, characterizing the effects of this kind of soft crowding agents is much more challenging. By incorporating an attractive interaction between enzymes and crowder particles in the dynamic energy landscape model, we revealed that non-specific protein-crowder attraction tends to compensate the entropic contribution and increase the enzyme activity compared to the inert crowder condition for the enzymes with the product release being the rate-limiting step (see Supplementary Fig. S3 and Supplementary Materials). For enzymes with closed-biased conformational propensity, such non-specific attractive interactions may even speed up the catalysis by promoting the product release step (Supplementary Fig. S4).

On the other hand, several studies have demonstrated that macromolecular crowding can significantly slow down substrate diffusion, leading to the inhibition of enzymatic catalysis[27,71-73]. This effect of crowding on substrate diffusion is neglected in our current model, in which substrate diffusion is assumed to be constant and much faster than domain motion. However, we anticipate that if substrate diffusion were dramatically slowed to become comparable to or even slower than domain motion, the crowding effect on substrate diffusion would become a non-negligible factor influencing catalysis. In particular, if substrate diffusion becomes the rate-limiting step, it would dominate the overall reaction kinetics and potentially overshadow the conformational dynamics effects reported in this study. This represents an important consideration for future investigations extending our framework to more extreme crowding conditions.

In summary, we investigated the impact of molecular crowding on the enzymatic catalysis by molecular simulations with a residue-resolved dynamic energy landscape model. Our computational results showed that adding inert crowding agent slows down the catalysis of the WT AdK. Further investigations into the AdK variant models with different conformational propensities revealed that molecular crowding can either enhance or diminish enzymic activity, depending on the nature of the rate-limiting step in the catalytic cycle. When the rate-limiting step involves the conformational closing and formation of the catalytically competent enzyme-substrate complex, molecular crowding enhances catalytic activity. Conversely, if the rate-limiting step involves the conformational opening of AdK and the product release, molecular crowding diminishes catalytic activity. This dual effect can be reconciled by considering the entropic contribution of molecular crowding to the increased stabilization of the closed conformation. We also investigated the effect of a soft crowding agent with attractive interactions on the catalytic activity. The results suggest that the attractive interactions of the soft crowder can compensate the entropic effect and enhance the enzymatic activity for the closed-biased conformational propensity compared to the inert crowding agent. These results provide a deeper understanding of how molecular crowding influences enzyme kinetics by modulating specific steps in the catalytic cycle. This insight has broader implications for enzymatic regulation in crowded cellular environments and highlights the importance of conformational dynamics in determining enzymatic efficiency under physiological conditions.

It is important to acknowledge that the crowding model used in this study is highly simplified compared with the physiological environment. In the simulations, crowders are represented as spherical particles with non-specific excluded-volume interactions. By contrast, intracellular crowding arises from a heterogeneous mixture of large biomacromolecules, including proteins, nucleic acids, polysaccharides, and their complexes, that have diverse sizes, shapes, and physicochemical properties. Such heterogeneity creates a much more complex environment than that captured by a homogeneous, hard-sphere approximation. Incorporating the explicit physicochemical features of realistic crowders will therefore be essential for

extending the dynamic energy-landscape framework to more accurately describe crowding effects on enzyme catalysis. As highlighted in previous studies, crowder size, polydispersity, and mixtures of different crowder species can exert significant and non-trivial effects on protein stability, conformational fluctuations, and binding equilibria. Moreover, crowding may influence substrate association kinetics and modulate the cooperative motions of multi-subunit enzymes. Nonetheless, the conclusions drawn from the minimal excluded-volume model presented here likely reflect an important underlying physical factor that persists even in more complex conditions.

In addition, the enzyme examined in this work undergoes substantial conformational transitions between open and closed states. The mechanisms identified here may therefore not directly generalize to enzymes that operate with negligible conformational rearrangements. Future extensions of the framework should explicitly incorporate coupling between crowders and the chemical step itself to achieve a more comprehensive understanding of enzyme catalysis under crowded conditions.

## Methods
### Dynamic energy landscape model of enzyme catalysis
In our prior studies, we developed a residue-resolved, structure-based AICG2+ model based on protein energy landscape theory and a multiscale strategy to investigate the allostery-coupled protein dynamics[65]. In this model, each amino acid is represented by a single bead centered on its $C_\alpha$ atom, yielding a total of 214 beads for AdK. The corresponding energy function is given by

$$
\begin{aligned}
V_{AICG}(\mathbf{r}, \mathbf{r_0}) = &V_{bond}^{loc}(\mathbf{r}, \mathbf{r_0}) + V_{angle}^{loc}(\mathbf{r}, \mathbf{r_0}) + V_{dihedral}^{loc}(\mathbf{r}, \mathbf{r_0}) \\
&+ V_{flp}^{loc}(\mathbf{r}) + V_{contact}^{nloc}(\mathbf{r}, \mathbf{r_0}) + V_{exv}^{nloc}(\mathbf{r})
\end{aligned}
\tag{2}
$$

Here, $\mathbf{r}$ denotes the coordinates of the coarse-grained residues in a given structure, and $\mathbf{r_0}$ refers to the reference coordinates from the native structure. $V_{bond}^{loc}$ enforces the covalent connectivity between consecutive residues via harmonic bonds. $V_{angle}^{loc}$ and $V_{dihedral}^{loc}$ impose structure-based restraints on bond angles and dihedral angles, respectively, using Gaussian potentials centered on native values. The flexible local potential term $V_{flp}^{loc}$ describes inherent sequence-dependent chain flexibility and secondary structure propensity of the protein chain, derived from a statistical survey of the coil library. The nonlocal terms, involving residue pairs separated by more than four amino acids in sequence, include $V_{contact}^{nloc}$, a structure-based attraction between residue pairs that form native contacts in the reference state, which shapes a funnel-like energy landscape biased toward the native conformation, and $V_{exv}^{nloc}$, a repulsive excluded-volume interaction that prevents steric clashes between other non-bonded residue pairs. Together, these terms produce a smooth, minimally frustrated energy landscape with a global minimum at the native state. More detailed description of the AICG2+ model can be found in previous works[65,74,75].

To model conformational transitions between distinct functional states, we employ a global interpolation scheme that smoothly merges two AICG2+ potential energy functions, each centered on a different reference structure (denoted as $\mathbf{r_1}$ and $\mathbf{r_2}$), as formulated in Eq. (3)[76]. This approach generates a multi-basin energy landscape that approximates the complex functional energy landscape of proteins and explicitly accounts for conformational interconversion. The interpolation introduces two key parameters: $\Delta V$, which controls the relative stability (energy gap) between the two basins, and $\Delta$, which modulates the height of the transition barrier. Both can be calibrated to match experimental data on state populations and transition kinetics. Using this scheme, we constructed a multiple-basin energy function for apo AdK by taking the experimentally resolved open (pdb code: 4AKE[77]) and closed (pdb code: 1AKE[78]) conformations as reference states. The main conformational changes in AdK involve opening and closing motions of the LID and NMP domains relative to the CORE domain. Accordingly, we introduced two separate double-basin energy terms to describe the LID-CORE and NMP-CORE motions, allowing the

two substrate-binding domains to move independently. In addition, a third double-basin energy term was added to describe interactions at the LID-NMP interface, capturing the coupling between the motions of the two domains. The total potential energy of the protein is therefore given by the sum of these three components. The movement of shared residues among these double-basin systems couples the motions of different domains. For the WT AdK, the $\Delta V_i$ values are optimized to reproduce the experimentally measured population of the closed state at apo condition ($P_{closed} = 0.3$). The optimization procedure started from an initial guess (e.g., $\Delta V_i = 0$). We then performed MD simulations of ligand-free AdK and calculated the resulting $P_{closed}$ value. If the computed $P_{closed}$ is higher than the experimental value, we increased the $\Delta V_i$ values to penalize the closed state; If $P_{closed}$ was below 0.3, we decreased the $\Delta V_i$ values. This iterative process continued until the simulated $P_{closed}$ converged to approximately 0.3. Using this calibrated framework, we can systematically adjust the $\Delta V_i$ parameters to generate computational models of AdK variants with predetermined conformational preferences. The energy function form and the parameter values of $V_{DB}^i$ can be found in Supplementary Materials (Supplementary Text and Table S1).

$$V_{DB}(r, r_1, r_2) = \frac{V_{AICG}(r, r_1) + V_{AICG}(r, r_2) + \Delta V}{2}$$
$$- \sqrt{\left(\frac{V_{AICG}(r, r_1) - V_{AICG}(r, r_2) - \Delta V}{2}\right)^2 + \Delta^2} \quad (3)$$

To avoid the sampling challenges inherent in explicit ligand binding simulations, we adopted an implicit treatment of ligands. In this approach, substrates and products are not represented as physical particles; instead, their binding effects are captured through ligand-specific, attractive interactions between residues lining the binding pockets. Specifically, successful ligand binding is modeled as the formation of native-like inter-residue contacts that stabilize the bound conformations[64]. The energetic contribution of ligand binding at site $ls_i$ is described by the potential:

$$V_{bind}^i(r, ls_i) = \sum_{IJ} \epsilon_i exp\left(-\frac{(r_{IJ} - r_{IJ}^0)^2}{2\sigma^2}\right) \quad (4)$$

where the summation spans residue pairs $(I, J)$ that would normally interact with the ligand if the ligand were explicitly included at binding site $ls_i$. Here, $r_{IJ}$ represents the instantaneous distance between residues $I$ and $J$ during simulation, while $r_{IJ}^0$ denotes the corresponding reference distance derived from the experimental substrate-bound structure. $\epsilon_i$ represents the ligand-specific interaction strength, and its values were calibrated against experimentally measured dissociation constants to ensure quantitative agreement with observed binding affinities[51]. Ligand binding and unbinding events are stochastically sampled using a Metropolis Monte Carlo scheme. Assuming a diffusion-limited binding process, the binding rate depends linearly on ligand concentration and is modulated by the geometric accessibility of the binding site:

$$k_{on} = f(S) k_{on}^0 [L], \quad (5)$$

where $[L]$ is the ligand concentration, $k_{on}^0$ is a baseline diffusion-limited rate constant, and $f(S) = 1/[1 + exp(-(S - S_0)/\sigma_S)]$ accounts for the accessibility of the binding pocket, quantified by its solvent-accessible surface area $S$. By appropriately choosing the parameters $S_0$ and $\sigma_S$, $f(S)$ ensures that binding occurs only when the pocket is sufficiently open and accessible[51]. Conversely, the dissociation rate incorporates the energetic penalty associated with breaking the ligand-stabilized contacts:

$$k_{off}^i = f(S) k_{off}^0 \, exp\left[-\frac{V_{bind}^i(r, ls_i)}{k_B T}\right], \quad (6)$$

where $V_{bind}^i(r, ls_i)$ is the current ligand-binding energy (as defined in Eq. 4) for the $i$th binding pocket, $k_B$ is the Boltzmann constant, and $T$ is the simulation temperature. At each Monte Carlo step (performed every 100 MD integration steps), the acceptance probability for a ligand binding or unbinding attempt is computed from the kinetic rates $k_{on}$ and $k_{off}$. This framework couples ligand exchange directly to both conformational state and ligand concentration, enabling realistic modeling of dynamic allostery during catalysis. Because ligands are treated implicitly, chemical conversion cannot occur through physical reaction coordinates. Instead, it is implemented via a kinetic Monte Carlo step, parameterized by experimentally determined forward ($k_f$) and reverse ($k_r$) catalytic rates[51]. This simplified treatment of the chemical step is physically justified for AdK, as experimental evidence indicates that product release, rather than chemistry itself, constitutes the rate-limiting step of the catalytic cycle[63]. Therefore, accurately capturing the conformational dynamics that govern substrate binding and product release represents the critical factor in modeling AdK's functional mechanism. Crucially, the chemical reaction is only allowed when the enzyme adopts a catalytically competent state—defined as the closed conformation with both ATP and AMP correctly bound and active-site residues properly preorganized. Both the ligand binding/unbinding and chemical reaction can modify the ligand binding states and therefore modulate the overall energy surface, reshaping the total energy landscape through changes in $V_{bind}$. Thus, the catalytic cycle emerges from the interplay between conformational dynamics on a given landscape and discrete transitions between landscapes driven by ligand and chemical state changes—yielding a dynamically evolving energy surface that guides functional progression. Further details regarding the model implementation, parameterization, and validation can be found in ref. 51.

**Modeling the molecular crowding environment**

For simplicity, we utilized a spherical bead with a radius of 8 Å to represent the crowder particle. This choice was made for two main reasons. First, at the same volume fraction, smaller crowders produce a stronger excluded-volume effect than larger macromolecules. Second, using uniform spherical crowders avoids complications from size differences and specific chemical interactions, allowing us to focus on the purely entropic effects of macromolecular crowding on protein conformational equilibria. Following the force field employed for crowding agents in prior studies[69,70], repulsive interactions were introduced both among crowder particles ($V_{cc}$) and between AdK and crowder particles ($V_{pc}$):

$$V_{cc} = \varepsilon_{cc}\left(\frac{\sigma_{ref}}{r_{IJ} - \sigma_{cc} + \sigma_{ref}}\right)^{12}$$
$$V_{pc} = \varepsilon_{pc}\left(\frac{\sigma_{ref}}{r_{IJ} - \sigma_{pc} + \sigma_{ref}}\right)^{12} \quad (7)$$

In Eq. (7), $r_{IJ}$ represents the distance between $I$-th and $J$-th beads. The strength of the repulsion potential is denoted by $\varepsilon_{cc} = 1.0$ kcal/mol and $\varepsilon_{pc} = 1.0$ kcal/mol. $\sigma_{ref} = 6$ Å serves as a reference value defining the range of the repulsive interaction. The crowder's diameter is given by $\sigma_{cc} = 16$ Å, while $\sigma_{pc} = 10$ Å is the sum of the crowder's radius and the protein amino acid. For comparison, we also tested the attractive interaction between the crowder particle and AdK (see Supplementary Materials for details).

In the simulation, the system was confined within a cubic box measuring 115 Å in each direction. A box potential was introduced to constrain the particles from moving beyond the box boundaries:

$$V_{box} = \begin{cases} 0, & d > 3\sigma \\ K_{box}\left(\frac{\sigma}{d}\right)^{12}, & 0.8\sigma \leq d \leq 3\sigma \\ K_{box}\left(\frac{\sigma}{0.8\sigma}\right)^{12}\left(1 + 12\frac{0.8\sigma - d}{0.8\sigma}\right), & d < 0.8\sigma \end{cases} \quad (8)$$

$K_{box} = 10$ kcal/mol is the strength of the box potential. $\sigma = 5$ Å represents the sharpness of the box wall. $d$ is the distance between the particle and the box wall. This results in a volume of 100 Å × 100 Å × 100 Å. Various numbers of crowder particles were introduced into the box to emulate different crowding volume fractions ($\phi$), as calculated by the following formula:

$$\phi = N \times \frac{4}{3}\pi R_c^3 / V \tag{9}$$

Here $N$ is the number of the crowder particles. $R_c = 8$ Å is the radius of the crowders. $V$ is the volume of the system. In this work, we performed simulations in crowding volume fractions of 0, 0.1, 0.2, 0.3, and 0.4, which contain 0, 47, 93, 140, and 186 crowder particles, respectively. It should be noted that we employed a soft confining (box) potential rather than periodic boundary conditions (PBC) to mimic a locally crowded and confined environment around the enzyme, whereas PBC effectively represents an infinite system. In all simulations, AdK was initially positioned near the center of the simulation box and surrounded by crowding agents. Due to the viscous, crowded environment—which strongly suppresses long-range diffusion—the enzyme rarely approached the box boundaries. Consequently, the confining potential exerted negligible influence on the enzyme's conformational dynamics or binding kinetics. By fitting the turnover rate against [ATP] using the Michaelis–Menten equation, we obtained the values for $k_{cat}$ and $K_M$, respectively. To investigate the influence of crowder size, we additionally performed simulations using larger crowders with a radius of 12 Å at $\phi = 0.3$, which corresponds to 51 particles in the same volume. It is important to note that both the 8 Å and 12 Å crowders are significantly smaller than typical cellular macromolecules. Nevertheless, these small inert particles generate a strong excluded-volume effect at high concentrations, allowing us to isolate entropic crowding contributions.

## Molecular simulations

All the simulations were carried out by our in-house modified version of CafeMol2.0[79], with the temperature of 300K controlled by a Langevin thermostat. The time step of the Langevin dynamics simulations was set to 0.2 $\tau$, with $\tau$ being the CafeMol time unit. Each trajectory lasted for $2 \times 10^8$ MD steps. No separate equilibration was performed, as our CG model reaches equilibrium rapidly under the simulation conditions. This exceptionally long simulation duration enables thorough sampling of the slow conformational transitions of AdK even under molecular crowding conditions (see Supplementary Fig. S5). Given the fact that our CG model reduces molecular complexity by simplifying degrees of freedom and energy functions, the time unit ($\tau$) does not directly correspond to physical time. To establish a meaningful kinetic timescale, we calibrated the simulation dynamics against experimental data by matching the rate of conformational transitions in the apo state of AdK in our previous works. Based on this calibration, $2 \times 10^8$ MD steps correspond to approximately 112 milliseconds of real time[51]. The transition rate for WT AdK in the apo state was determined to be ~7.0 ms$^{-1}$ [67]. The turnover rate was calculated from 20 independent replica trajectories. ADP concentration was maintained at 0 in all simulations. MDTraj and PyMOL were utilized for all analyses and structural visualization[80,81]. It is important to emphasize that while the absolute turnover rates depend on the specific experimental reference used for this timescale mapping, such calibration does not affect the relative trends across different crowding conditions. Therefore, all qualitative and comparative conclusions regarding catalytic efficiency, allostery, or conformational equilibria remain robust.

## Data analysis

The turnover rate was calculated from the mean time for a complete AdK catalytic cycle. Kinetic parameters, $k_{cat}$ and $K_M$, were derived by fitting the calculated turnover rates to the Michaelis–Menten equation. To elucidate rate-limiting steps under crowding, we computed the MFPTs for forming the catalytically competent state and for substrate release following previous work[59]. Uncertainties in MFPT values were quantified using bootstrap analysis. Results were visualized with UltraPlot[82].

## Data availability

The key simulation data used to generate the figures in this manuscript, along with example input files, initial structures, and representative molecular dynamics trajectory files, are available at https://github.com/wtren/crowding_adk_2025.

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

## Acknowledgements

The authors thank Gilad Haran for insightful discussions. This work was supported by the Basic Research Program of Jiangsu Province (BK20253050), the National Natural Science Foundation of China (Nos. 11974173, 12090052, and 12347102) and the grant from Wenzhou Institute, University of Chinese Academy of Sciences (WIUCASQD2021010, WIUCASQD2022019, WIUCASQD2023015). The authors also thank the support from HPC Center of Wenzhou Institute, University of Chinese Academy of Sciences, HPC center of Nanjing University, e-Science center of Nanjing University, Nanjing Kunpeng&Ascend Center of Cultivation, and the Nanjing Key Laboratory for Cardiovascular Information and Health Engineering Medicine (funded by the Nanjing Municipal Health Commission) and its Jiangsu counterpart.

## Author contributions

W.R., Y.Z., W.L., and W.W. designed the research, W.R. and J.L. performed MD simulation, W.R, J.L., H.H., J.Z., and Y.C. analyzed data, W.R., Y.Z., W.L., and W.W. wrote the paper.

## Competing interests

The authors declare no competing interests.
