## [Transparent Peer Review file · Communications Chemistry]

Allosteric Regulation of Enzymatic Catalysis by Molecular Crowding

Corresponding Author: Dr Weitong Ren

Version 0:

Reviewer comments:

Reviewer #1

(Remarks to the Author)

Ren et al investigate enzymatic reactions in crowded environments. This is an important topic. Unlike macromolecular diffusivity, reactions have been much less investigated under physiologically relevant crowding conditions. This work may contribute to our understanding of how enzymatic reactions proceed in crowded environments. However, I have several comments and suggestions that the authors should consider in the manuscript before making any recommendations.

Overall, the manuscript is quite difficult to read. In particular, the method is not clearly explained, and its novelty is unclear. It appears the authors use an approach that was already published (ref 48); however, in the abstract, they claim to provide "a computational framework to understand the diversity of existing experimental observations to the molecular crowding effects on enzyme catalysis." This must be corrected, and, as I said, the method must be much better explained than it is at the moment.

Concerning a crowded environment. It is unclear why the authors mention "liquid-liquid phase separation" (which they do not consider explicitly anyway) to justify crowding conditions, as such conditions are ubiquitous in the cell interior. For example, in the E. Coli cytoplasm, macromolecules occupy about 40 percent of the cell volume (see e.g., [https://doi.org/10.1016/0022-2836\(91\)90499-V](https://doi.org/10.1016/0022-2836(91)90499-V), <https://doi.org/10.1091/mbc.e12-08-0617>). Yet, the authors consider monodisperse spherical crowders of a size of 8 Å. These crowders are extremely small compared to physiological conditions. Moreover, in living cells, crowding is typically due to macromolecules of various sizes, from several to tens of nanometers. While the authors acknowledge this size dispersity in the manuscript, I would suggest providing an additional discussion of how these more realistic conditions would modify the results of the present work. In particular, the choice of the crowder size should be more carefully discussed as it may quite significantly influence the results.

The authors should also be careful with the terminology "mutant" enzymes. If I understand correctly, it is not a mutant enzyme, but the wild-type adenylate kinase with modified (in a computer only) opening-closing probabilities. While this is interesting to investigate, saying it is a "mutant enzyme" might be misleading, as no mutation has actually been done, and it remains unclear if it can be done experimentally.

Further comments:

- In the introduction, the authors write that "Over the past two decades, experimental studies have investigated enzyme activity under crowded conditions." However, there are also several theoretical/simulation studies of reactivities under crowding conditions, which the authors do not mention at all (see e.g. <https://doi.org/10.1021/ja060483s>, <https://doi.org/10.1021/jz900023w>, <https://doi.org/10.1038/s41598-018-37034-3>, <https://doi.org/10.1371/journal.pcbi.1000833>, <https://doi.org/10.1039/D0CP06631A>, <https://doi.org/10.1103/PhysRevLett.130.258401> but there are more, particularly seminal Minton works). The authors should mention and discuss these works in the manuscript, possibly comparing them with their own approach.

- I suggest providing the scheme of how the enzymatic reaction proceeds, and explaining the reaction more clearly, for instance, providing information as to whether the enzyme closes upon binding, if the binding affects the closing probability, etc, and what computational assumptions are compared to the actual reaction mechanism.

- The paragraph below Eq. (1) is unclear and must be significantly improved. For instance, this sentence does not seem to be complete: "Because AdK has two binding pockets, and each binding site can bind substrate, product, or remain unoccupied, leading to maximally nine possible binding states." But as I said, the whole paragraph needs improvement.
- The authors write that "the non-monotonic relationship between KM and crowding volume fraction is likely associated with the modulation of the relative stability of the closed form of AdK by crowding, which will be discussed in the later subsection." While the authors do discuss the stability of the closed form in relation to crowding, they do not relate it to the non-monotonic KM behaviour. Furthermore, the authors write "in a moderately crowded environment, an increased presence of crowding agents enhances the stability of the closed conformation of AdK, resulting in a higher affinity to substrates." I agree that crowding enhances the stability of the closed form, but why does it result in a higher affinity to substrates?
- Fig. 2: Why are the error bars huge for KM but small for kcat? How were the error bars actually computed?
- The authors write: "energy gap ΔV was optimized according to experimentally measured P_{closed} ." I suggest providing more details on how it was optimized.
- Further, the authors state that "upon introducing crowder particles, the enzymatic activity monotonically diminishes with P_{closed} ." However, Fig. 3B shows a small peak at low P_{closed} . In Fig. 3A, not all curves seem to be in saturation, raising the question of how accurate the fitting was.
- Fig. 4 shows that for $P_{\text{closed}} = 0.31$ and 0.54 , there is a non-monotonic behaviour versus occupied volume fraction. I find this quite interesting and worth discussing in more detail in the manuscript.
- The authors write that crowding "does not affect the relative population of the two pathways," but it is not substantiated, and it is difficult to judge based solely on Fig. 5A.
- The authors state, "The acceleration of the substrate binding step primarily arises from the crowding enhanced open-to-closed conformational transition." I am unsure if I understand this. Crowding, quite generally, stabilises the closed conformation. What does it mean that it enhances a transition? Quite generally too, crowding also enhances molecular binding (see e.g. <https://doi.org/10.1242/jcs.03063>), so perhaps this is the reason for the accelerated binding step?
- In Fig. 6, there is no A, B, etc. Also, the figure caption is unclear (e.g., "starting from the open state for the product release")
- The authors conclude that "adding crowding leads to opposite effects on the overall catalytic kinetics for the open-biased AdK model and closed-biased AdK model." As I already mentioned, Fig. 6 indicates a non-monotonic behaviour, which I find even more interesting and suggest discussing. In the regime where crowding reduces the activity, Skora et al provided an analytical equation for the catalytic activity as a function of crowding (<https://doi.org/10.1039/D0CP06631A>). It would be interesting to see if this is consistent/fits to the authors' results.
- The authors should explain the meaning of a vertical line in $\langle r|r_0 \rangle$ in Eq. 2. They should also more clearly explain the model in this equation. For instance, how $V_{\text{exc}}^{\text{non-local}}$ is non-local if it depends solely on positions r ? The same concerns other terms.
- The authors further write that the apstate is decomposed into "3 double-basin systems describing the conformational motions between LID-CORE domains, NMP-CORE domains and LID-NMP domains." What does it mean: "a motion between the domains"? Motion of what? This must be better explained. Also, is this potential actually additive? I would suppose conformational changes in one domain can affect the changes in the other domain.
- Concerning Eq. (3), do I understand correctly that it means adding a binding energy between residuals of the enzyme that would be in contact with a ligand if the ligand were explicitly modelled? This piece must be clarified.
- The simulation method (as I already mentioned) must be more clearly explained. For instance, regarding Eq. (4), do I understand correctly that with the probability following from Eq. (4), some residuals of the enzyme that would face a ligand (if it were explicitly modeled) become bound with the potential given by Eq. (3)? Why this particular form of Eq. (4)? Similarly, what motivates the choice of the expression for koff given by Eq. (5)? One does not have to provide derivations but giving some clear physical reasoning would be very helpful.
- If I am not mistaken, upon successful MC steps describing ligand binding, the interaction potentials are dynamically changed. Doesn't it generate numerical instabilities in MD simulations due to large forces (potentially emerging because the system is not "equilibrated" for this potential)?
- The authors impose a box potential on their system. Why not use periodic boundary conditions? Does this box potential affect the enzyme kinetics?
- Providing some more details on MD simulations would also be helpful, e.g., on the CafeMol time units. I am somewhat confused how the authors managed to reach 112 milliseconds. With how many beads was the enzyme represented?
- In Supplementary materials, R in Eq.(1) is not defined. Also, where does the form of Eq. (1) follow from? Similarly, I suggest providing more information on the choice of the reaction coordinate, Eq. (3), in the SM.

- The authors write: "Three double-basin energy terms are employed to describe the interaction of each pair of two domains of AdK." If there are two domains, there is only a single pair.
- This sentence in the SM is not fully clear: "When a substrate or product was thought to bind to AdK, bunches of ligand-mediated contacts around the ligand binding site were added to the AICG2+ energy function to stabilize the closed conformation."
- Eq. (4) in the SM, V is the extra energy contributed by the ligand-mediated contacts. What is this extra energy? Is it a function? Is it a value? Why in Eq. (3) and (4) in the SM, there is a proportionality sign, while the same equations in the main text have equality. Seeing the difference, it would nevertheless be good to keep consistency between the SM and the main text.
- On page 3 of the SM, the authors refer to figure S1, but it does not seem to be the correct figure.
- Various parameters in Eq. (5) of the SM are also not defined.

Reviewer #2

(Remarks to the Author)

This manuscript addresses the timely and complex question of how molecular crowding influences enzymatic catalysis, a topic of broad significance in biophysics. The authors employ adenylate kinase (AdK), a model system extensively studied in both experimental and computational contexts, to explore how inert crowders reshape the conformational equilibrium and impact individual steps of the enzymatic cycle. Using coarse-grained simulations, they provide quantitative insights into how crowding stabilizes particular conformational states and thereby modulates turnover, substrate binding, and product release. Importantly, the study reveals that crowding can either enhance or inhibit activity depending on the rate-limiting step of the cycle, offering a mechanistic framework that reconciles diverse prior observations. Overall, this is a carefully executed and conceptually significant work that will be of interest to a broad readership. However, there are several important issues—mainly concerning the clarity and presentation of the manuscript—that the authors should address before it can be considered for publication. I expect that resolving these points should be relatively straightforward.

Major comments

1. There appears to be some confusion regarding the definition of $P(\text{closed})$. Initially, it is defined as the relative population of the closed conformation. Later in the manuscript, in particular in Figure 4, this definition seems inverted, with high values corresponding to an open-biased conformation. In addition, the trends in Figures 3 and 4 appear to be opposite: while in Figure 3 adding crowder at low $P(\text{closed})$ enhances activity, in Figure 4 the reverse seems to be the case. A low $P(\text{closed})$ is also designated as closed-biased in the cartoon, suggesting that the panels in Figure 4 may have been accidentally swapped. Similarly, in line 206 conformations with $P > 0.3$ are described as open-biased, where closed-biased would be correct. Clarifying the definition of $P(\text{closed})$ and checking the consistency of figures and text would greatly improve readability.
2. The authors investigated not only inert crowders but also attractive interactions. However, these results are not discussed in the RESULTS section and can only be found in the SI, despite being referred to multiple times in the manuscript and discussed in detail in the DISCUSSION. These data could provide an interesting link to experimental findings mentioned by the authors, such as the activity enhancement by urea, which is notable given urea's typical role as a denaturant. I recommend that the authors include at least a brief summary of the main results for attractive crowders in the RESULTS section.
3. Figure 6 lacks panel labels (A, B, etc.), which makes it difficult to follow the discussion in the text. Furthermore, the authors state: "The acceleration of the substrate binding step primarily arises from the crowding enhanced open-to-closed conformational transition" (lines 252–253), but the figure does not actually show the conformational transitions. Also, it would also be helpful to refer to Figure 6 earlier in the text, no later than line 250.
4. The authors consider $P(\text{closed})$ as a critical determinant of activity. However, for equilibrium populations to be the controlling factor, the conformational transitions themselves must be fast and not rate-limiting. The authors should clarify this prerequisite in the text.
5. The reported changes in K_M are small (60 to 70 μM) and fall within the error bars (lines 165–171). I would therefore consider K_M to be unaffected by crowding, in contrast to the clear changes observed in k_{cat} . From a theoretical standpoint, the only mechanism by which K_M might change is if the various substrate-bound species are affected differently by crowding (see next point), thus manipulating substrate binding and dissociation.
6. The conformational equilibrium of AdK is known to be strongly influenced by substrate binding. The authors mention nine different substrate-bound species, but do not discuss whether crowding has distinct effects on these species. Among other places, this could be relevant for the discussion in lines 171–175: while crowding may hinder substrate binding by favoring the closed state in the apo form, it could also strengthen ligand binding if primarily the substrate-bound species are stabilized.

Minor comments

1. Line 22: Please provide a reference for the specific number (30%).
2. Figure legend 1: In panel 1B, were the substrates modeled into the crystal structure of the protein? Otherwise, the shown molecule would be AP5. Second, the sentence "AMP can nonspecifically bind at the AMP site but with lower affinity" is unclear; I assume the ATP site is meant. Experimental evidence argues against competition between ATP and AMP at the ATP site. Since AMP binding at the ATP site is not considered later in the paper, I recommend removing this sentence.
3. Line 183: Please provide a reference for the value of 0.3.

4. Line 196–197: “In dilute solution, the k_{cat} value exhibits a biphasic trend. It increases when $P_{closed} < 0.05$ and decreases when $P_{closed} > 0.05$.” It would be clearer to state that k_{cat} is maximal at $P_{closed} \sim 0.5$.
5. Line 205: “decreases” would be correct instead of “increases.”
6. Figure legend 3: I assume that all lines, not only the solid ones, correspond to MM fits? Please clarify.
7. Line 233: It would be helpful to mention the substrate concentrations in the text, not only in the figure legend.
8. Lines 243–244: The statement “Notably, more substantial enhancement is observed for the AdK variants with moderate P_{closed} values (e.g., 0.3)” is expected, since the energy difference between the states is lowest at moderate P values.
9. Line 250: From Figure 6, product release appears to be “insensitive” rather than “less sensitive” to substrate concentration.
10. Lines 316–337: The phrase “Consistent with experimental observation” should be clarified. The experimental references cited in the Introduction mainly concern shifting the conformational equilibrium by mutations or other means, not by inert crowding. A specific reference should be provided, or the sentence adjusted.
11. Figure 7: In light of major comment 6, the authors might consider splitting the first step into substrate binding and conformational transition. Faster closing does not promote turnover in the apo protein, as it can hinder substrate binding, but once substrates are bound, faster closing accelerates the reaction.
12. SI page 3: The reference in the sentence “This classification was employed to characterize the kinetics of AdK’s functional dynamics, encompassing processes such as substrate binding, product release, and the catalytic cycle (see Fig S1)” is misleading, as Fig. S1 does not show this. Please correct the figure reference.
13. Overall, the message of the manuscript is clear and easy to follow. However, several typographical and minor errors slightly impair readability. I recommend a careful proofreading to correct these.

Version 1:

Reviewer comments:

Reviewer #1

(Remarks to the Author)

The authors have quite carefully addressed the referees' comments. I have only a few comments that I suggest addressing before publication.

- The authors write that “a key advance” of their work is the integration of “the enzyme model with a macromolecular crowding model”. This appears somewhat straightforward to me, however. In my opinion, the main advance is the investigation of crowding effects on complex enzymatic reactions, which are still rare in the literature.

- The authors write that “moderate crowding shifts the conformational equilibrium toward this closed state, which can enhance effective substrate affinity” (also in the main text: “Moderate crowding stabilises the closed conformation and accelerates substrate binding”). Why does stabilising closed configurations accelerate substrate binding? Or why would the intrinsic affinity increase? If the probability of the closed conformation increases, one could actually expect the opposite (as the authors write for strong crowding conditions). Perhaps closing stabilises the bound state (hindering dissociation), but only provided a substrate does bind to the active site, and through this stabilization enhances reactivity? For stronger crowding, the binding is of course hindered, per se. These points should be clarified.

- In some cases, the authors provide the response, but it is not taken into account in the manuscript. For instance, regarding the error bars, the authors added in the captions that they are based on bootstrapping. However, perhaps I missed it, but I do not see any explanation in the manuscript for the large error bars for KM. This might be useful to readers. Similarly, the impact of the confining box on enzyme kinetics, particularly the fact that the enzyme does not approach the box boundaries within typical simulation time scales, is also important.

- Regarding three double-basin systems, I assume this representation is an approximation, so I suggest stating this explicitly in the manuscript.

- The authors avoid explicit simulations of ligands. As I understand, this is a trick simplifying and speeding up the simulations. However, ligand diffusivity can also be affected by crowding, which would influence overall enzymatic kinetics. It seems this effect is not taken into account within the authors' approach. Some studies indicate quite a significant slowdown of even small molecules (see <https://doi.org/10.1002/jmr.709>, <https://doi.org/10.1021/acs.nanolett.3c05100>, <https://doi.org/10.7554/eLife.19274>). How would that effect impact the authors' results? I suggest commenting on this point in the manuscript.

Reviewer #2

(Remarks to the Author)

The authors have addressed all my concerns with great effort and care, and I recommend the publication of their work. I have two minor recommendations the authors might consider for the final manuscript.

1) I would appreciate it if explicit numbers for the transition rates were given, for an easier comparison with the explicit numbers for turnover. Line 561 would be a good spot, in my opinion. Also including units in Table S1 would be better.

2) The revised Fig. 7 is inaccurate, as it suggests that closing might occur before substrate binding. I recommend removing the arrow between the non-substrate-bound closed conformation and the bound closed conformation.

Responses and the corresponding changes

We thank the reviewers for their invaluable comments and suggestions, which have substantively helped strengthen our manuscript. In the revised manuscript, we have adequately addressed the all the questions and concerns. The point-by-point responses and the corresponding changes are listed below. The changes are marked in blue in the revised manuscript.

Reviewer 1

"Ren et al investigate enzymatic reactions in crowded environments. This is an important topic. Unlike macromolecular diffusivity, reactions have been much less investigated under physiologically relevant crowding conditions. This work may contribute to our understanding of how enzymatic reactions proceed in crowded environments. However, I have several comments and suggestions that the authors should consider in the manuscript before making any recommendations.

Overall, the manuscript is quite difficult to read. In particular, the method is not clearly explained, and its novelty is unclear. It appears the authors use an approach that was already published (ref 48); however, in the abstract, they claim to provide "a computational framework to understand the diversity of existing experimental observations to the molecular crowding effects on enzyme catalysis." This must be corrected, and, as I said, the method must be much better explained than it is at the moment."

Response:

We sincerely thank the reviewer for the constructive comments regarding the clarity and novelty of our methodology. We acknowledge that the original manuscript did not sufficiently explain the computational framework and its improvements compared to prior approaches, including the method referenced in Ref. 48. In the revised manuscript, we have substantially expanded the Methods section to provide a comprehensive description of our modeling approach, including the underlying assumptions, the implementation of the coarse-grained crowding environment, and how enzyme dynamics are coupled to the catalytic cycle. The novelty of the method used in this work compared to previous work was clarified.

While the computational model used in this work builds upon the enzyme model developed

in our earlier work, the key advance lies in **integrating the enzyme model with a macromolecular crowding model to systematically explore how crowding environment modulates catalytic dynamics**. This combined framework enables us to perform molecular simulation for the fully enzymatic cycle under crowding environment, which is a challenging task for existing approaches.

We have revised the Abstract and Introduction accordingly to more accurately reflect the scope and contribution of this work. We believe these changes significantly improve the readability, transparency, and scientific rigor of the manuscript.

"Concerning a crowded environment. It is unclear why the authors mention "liquid-liquid phase separation" (which they do not consider explicitly anyway) to justify crowding conditions, as such conditions are ubiquitous in the cell interior. For example, in the E. Coli cytoplasm, macromolecules occupy about 40 percent of the cell volume (see e.g., [https://doi.org/10.1016/0022-2836\(91\)90499-V](https://doi.org/10.1016/0022-2836(91)90499-V), <https://doi.org/10.1091/mbc.e12-08-0617>). Yet, the authors consider monodisperse spherical crowders of a size of 8 Å. These crowders are extremely small compared to physiological conditions. Moreover, in living cells, crowding is typically due to macromolecules of various sizes, from several to tens of nanometers. While the authors acknowledge this size dispersity in the manuscript, I would suggest providing an additional discussion of how these more realistic conditions would modify the results of the present work. In particular, the choice of the crowder size should be more carefully discussed as it may quite significantly influence the results."

Responses:

We sincerely thank the reviewer for the insightful comments and suggestions regarding the physiological relevance of our crowding model and the effect of crowder size.

We fully agree with the reviewer that macromolecular crowding in living cells is ubiquitous even without involving liquid-liquid phase separation. By referencing LLPS, we aimed to **emphasize that cellular crowding is not just a passive background condition in cellular environment. Instead, cells actively use phase separation to concentrate specific molecules into condensates, dynamically regulating crowding to support biological functions**. In this work, we improved the related discussion of LLPS to clarify the above points.

As also commented by the reviewer, macromolecular crowding in living cells involves a

broad distribution of molecular sizes ranging from several to tens of nanometers. our current model employs monodisperse spherical crowders of 8 Å in radius, which is indeed much smaller than typical physiological crowders. In this work, the choice of a small, uniform crowder size was motivated by the following considerations: 1) Small crowders generate a stronger excluded-volume effect compared to larger particles; 2) A uniform crowder size improves computational tractability and allows us to isolate the entropic effects of macromolecular crowding in a controlled manner, without confounding factors such as size polydispersity or specific chemical interactions. Moreover, this simplified crowder model has been used in prior studies to investigate crowding effects on protein folding and functional conformational transition [Biophys J. 2010 Jan 20;98(2):315–320, J. Chem. Theory Comput. 2020, 16, 2, 1319–1332, JACS Au 2025, 5, 10, 4916–4935]. Following the reviewer’s suggestion, we have performed additional simulations with larger crowders (radius = 12 Å) at equivalent volume fractions. As expected, these larger crowders produce weaker excluded-volume effects on the protein’s conformational equilibrium (Figure R1), confirming that crowder size significantly modulates crowding effects. These new results are now included in the revised manuscript and supporting materials.

We fully acknowledge that our simplified crowding model is unlikely to capture the full complexity of heterogeneous intracellular environments. In this revised manuscript, we have explicitly addressed this limitation in the Discussion section, emphasizing that our primary aim is to elucidate fundamental physical principles of crowding rather than to reproduce quantitatively the specific conditions within living cells. We appreciate the reviewer’s comment, which has strengthened our manuscript by prompting us to better contextualize both the capabilities and limitations of our approach.

Figure R1. Effect of molecular crowding on the catalytic activity of WT AdK at $\phi =$

0.3 using crowders of different size.

Corresponding changes to the manuscript:

1. (page 2, line 29) The following sentences were added in the main text of this revised manuscript: "Beyond this baseline level of heterogeneous crowding in the cytosol, cells actively regulate and organize crowding effects through higher-order spatial and functional compartmentalization via the formation of biomolecular condensates. These condensates represent dynamically organized, highly concentrated assemblies that impose structural organization, selectively control molecular composition and mobility, and thereby fine-tune biochemical reaction rates, signaling pathways, and enzymatic activities."

2. (page 17, line 314) The following sentences were added in the main text of this revised manuscript: "As noted earlier,... To further clarify these contributions, we performed additional simulations of WT AdK under crowded conditions at $\phi=0.3$ using larger crowder particles ($r=12\text{\AA}$). Interestingly, these larger crowders produced a weaker excluded-volume effect (Fig. S2), consistent with the expectation that larger crowders suppress enzymatic activity less strongly than smaller ones."

3. (page 17, line 334) We added the following text in the revised manuscript. "We emphasize that these additional simulations are intended to illustrate the rich and nontrivial nature of crowding effects on enzyme dynamics, rather than to provide a comprehensive characterization of all possible crowding regimes"

4. (page 25, line 520) We added the following statement to the revised manuscript. "This choice was made for two main reasons... Second, using uniform spherical crowders avoids complications from size differences and specific chemical interactions, allowing us to focus on the purely entropic effects of macromolecular crowding on protein conformational equilibria."

"The authors should also be careful with the terminology "mutant" enzymes. If I understand correctly, it is not a mutant enzyme, but the wild-type adenylate kinase with modified (in a computer only) opening-closing probabilities. While this is interesting to investigate, saying it is a "mutant enzyme" might be misleading, as no mutation has actually been done, and it remains unclear if it can be done experimentally."

Response:

We thank the reviewer for this important clarification. As commented by the reviewer, our simulations do not involve real amino acid substitutions; rather, we computationally modulate parameters that govern the relative stability of the closed conformation to mimic the energetic effects that certain mutations might induce. This approach allows us to explore how shifts in the conformational free energy landscape affects catalytic dynamics. Indeed, prior experimental work has demonstrated that specific point mutations in AdK (e.g., in the LID or NMP domains) can significantly reshape its energy landscape, alter domain motions, and impact catalytic efficiency [J. Am. Chem. Soc. 2012, 134, 16562–16570, Nature 2018, 558, 324–328]. We agree with the reviewer that referring to it as a “mutant enzyme” could be misleading as our current model is a *in silico* representation of such perturbations and not a true mutant. In this revised manuscript, we have rephrased the related terminology as “variant” and discussion to clarify that these are AdK variants, and we explicitly stated that no physical mutations were introduced.

Corresponding changes to the manuscript:

(page 10, line 214) The following sentence was added in the revised manuscript “It is important to note that this approach creates theoretical models that mimic the energetic effects of mutations without involving actual amino acid changes, and do not directly correspond to real AdK mutants.”

“- In the introduction, the authors write that “Over the past two decades, experimental studies have investigated enzyme activity under crowded conditions.” However, there are also several theoretical/simulation studies of reactivities under crowding conditions, which the authors do not mention at all (see e.g. <https://doi.org/10.1021/ja060483s>, <https://doi.org/10.1021/jz900023w>, <https://doi.org/10.1038/s41598-018-37034-3>, <https://doi.org/10.1371/journal.pcbi.1000833>, <https://doi.org/10.1039/D0CP06631A>, <https://doi.org/10.1103/PhysRevLett.130.258401> but there are more, particularly seminal Minton works). The authors should mention and discuss these works in the manuscript, possibly comparing them with their own approach.”

Response:

We sincerely thank the reviewer for highlighting this important omission. In the revised manuscript, we have expanded the Introduction sections to include a more comprehensive overview of these key computational studies, including the seminal works by Minton and the references kindly provided by the reviewer (e.g., J. Am. Chem. Soc. 2006, 128, 7980;

J. Phys. Chem. Lett. 2010, 1, 23; Sci. Rep. 2019, 9, 1263; PLoS Comput. Biol. 2010, 6, e1000833; Phys. Chem. Chem. Phys. 2021, 23, 11829; Phys. Rev. Lett. 2023, 130, 258401). We now explicitly discuss how these prior approaches model crowding effects and compare their methodologies and findings with our multiscale framework.

Corresponding changes to the manuscript:

1. (page 3, line 62) The following sentences were added in the main text of this revised manuscript: “Complementary theoretical and computational studies further underscore the complexity of crowding effects on enzymatic catalysis. ... Taken together, the diverse and sometimes opposing effects of molecular crowding on enzyme activity underscore the difficulty of achieving a unified mechanistic understanding and highlight the need for integrative approaches that explicitly account for the interplay between enzyme dynamics, reaction kinetics, and the crowded cellular environment.”

- I suggest providing the scheme of how the enzymatic reaction proceeds, and explaining the reaction more clearly, for instance, providing information as to whether the enzyme closes upon binding, if the binding affects the closing probability, etc, and what computational assumptions are compared to the actual reaction mechanism.

Response:

We thank the reviewer for this constructive suggestion. In the revised manuscript, we added a schematic cartoon (new Figure 1C) that clearly illustrates the catalytic cycle of AdK, including substrate binding, open-to-closed conformational transition, phosphoryl transfer, and product release. The figure explicitly illustrates that AdK preferentially adopts an open conformation in the absence of substrates and undergoes domain closure upon ATP and AMP binding, a step that brings the substrates into proximity for catalysis.

Furthermore, in the Introduction and Methods section, we now clarify key computational assumptions underlying our model: (i) AdK preexists as an equilibrium between open and closed conformations even in the absence of substrates, with the open state being thermodynamically dominant [PNAS. 2007, 104 (47) 18496-18501; J Am Chem Soc. 2012 ;134(40):16562-70]; (ii) Substrate binding shifts this equilibrium by stabilizing the closed conformation through favorable interactions that reshape the underlying free energy landscape [J Am Chem Soc. 2012 ;134(40):16562-70]; (iii) The chemical step (phosphoryl transfer) is assumed to occur rapidly once the catalytically competent closed state is formed, allowing us to treat turnover as limited by conformational dynamics and substrate

binding rather than by the chemistry itself based on previous studies [PNAS. 2009, 106, 17359-17364]. We also discuss how these assumptions align with experimental and computational insights into the enzymatic mechanism of AdK. In addition, we explicitly depict the changes in the free-energy landscape of AdK during catalysis in Fig. 1C using a schematic representation, in order to visually convey the underlying assumptions of our model. This clarification ensures transparency about the scope, approximations, and limitations of our simulation approach, while highlighting its suitability for probing crowding-induced modulation of conformational equilibria and ligand-binding kinetics.

Corresponding changes to the manuscript:

1. (page 4, line 81) The following sentences were added in the main text of this revised manuscript: "In the apo state (without substrates and products), the enzyme preexists as an equilibrium between open and closed conformations even in the absence of substrates (Fig. 1A, B), with the open state being thermodynamically dominant. Substrate binding to their respective sites shifts this pre-existing equilibrium by stabilizing the closed conformation through favorable interactions that reshape the free energy landscape, forming a catalytically competent state for phosphate transfer. After the phosphoryl transfer reaction occurs in the closed conformation, the enzyme reopens to release the products"
2. (page 7, line 132) The following sentences were added in the main text of this revised manuscript: "The framework is based on the conformational selection (or population shift) mechanism, which posits that the enzyme intrinsically samples both open and closed conformational states even in the absence of ligands. Ligand binding selectively stabilizes pre-existing closed states, thereby shifting the conformational equilibrium toward the catalytically competent ensemble. The energy function comprises two components that jointly encode how ligand occupancy modulates the enzyme's conformational landscape"
3. (page 25, line 503) The following sentences were added in the main text of this revised manuscript: "This simplified treatment of the chemical step is physically justified for AdK, as experimental evidence indicates that product release, rather than chemistry itself, constitutes the rate-limiting step of the catalytic cycle⁶⁶. Therefore, accurately capturing the conformational dynamics that govern substrate binding and product release represents the critical factor in modeling AdK's functional mechanism"

- The paragraph below Eq. (1) is unclear and must be significantly improved. For instance, this sentence does not seem to be complete: "Because AdK has two binding pockets, and

each binding site can bind substrate, product, or remain unoccupied, leading to maximally nine possible binding states." But as I said, the whole paragraph needs improvement.

Response:

We thank the reviewer for this valuable feedback. In the revised manuscript, we improved this section to improve its logical flow and clarity. Specifically, we now explicitly explain that AdK possesses two distinct binding pockets—one for ATP (or ADP) in the LID domain and one for AMP (or ADP) in the NMP domain. Since each site can be occupied by a substrate, a product, or remain empty, the system can theoretically populate up to $3 \times 3 = 9$ distinct binding states. We clarify that not all of these states are equally probable or catalytically relevant, and we describe how our model accounts for the occupancy-dependent conformational equilibria (e.g., open vs. closed states) and their influence on the reaction flux. The revised paragraph now provides a coherent foundation for Eq. (1) and better integrates with the framework introduced in the following section.

Corresponding changes to the manuscript:

1. (page 8, line 149) The following sentences were added in the main text of this revised manuscript: "AdK possesses two distinct binding pockets: one specific for the AMP or ADP and the other for the ATP or ADP.... Changes in ligand state, through binding, unbinding, or chemical reaction, dynamically reshape the underlying energy landscape, thereby driving the progression of the catalytic cycle."

- The authors write that "the non-monotonic relationship between K_M and crowding volume fraction is likely associated with the modulation of the relative stability of the closed form of AdK by crowding, which will be discussed in the later subsection." While the authors do discuss the stability of the closed form in relation to crowding, they do not relate it to the non-monotonic K_M behaviour. Furthermore, the authors write "in a moderately crowded environment, an increased presence of crowding agents enhances the stability of the closed conformation of AdK, resulting in a higher affinity to substrates." I agree that crowding enhances the stability of the closed form, but why does it result in a higher affinity to substrates?

Response:

We thank the reviewer for this helpful comment. The slight decrease in K_M under moderate crowding (Fig 2C) can be explained by stabilization of the closed conformation. Previous experimental and computational studies have shown that the closed state of AdK exhibits

stronger substrate binding due to the formation of more protein–substrate contacts in this conformation (J. Am. Chem. Soc. 2012, 134, 16562–16570; PNAS 2017, 114, (24), 6298–6303). Moderate crowding shifts the conformational equilibrium toward this closed state, which can enhance effective substrate affinity. In contrast, high crowding over-stabilizes the closed state, which hinders the formation of catalytically competent (productive substrate-bound) states while promoting non-productive ligand-bound states, leading to an increase in K_M . We computed the formation probability of the productive substrate-bound state at various crowding conditions at $[ATP] = 9.5 \mu\text{M}$, and the trend is consistent with the observed changes in K_M . However, because the observed changes in K_M are small and fall within the fitting uncertainty, these results should be interpreted with caution.

Corresponding changes to the manuscript:

(page 9, line 194) The original sentences “Intriguingly, we observed a non-monotonic relationship between K_M and the crowding volume fraction (see Fig. 2C)...” is replaced by following sentences in the main text of this revised manuscript: “In contrast, K_M shows a slight decrease under low and moderate crowding,.... However, the fitted K_M values show substantial uncertainty, and the observed changes are small; therefore, these trends should be interpreted with caution and not overemphasized.”

- Fig. 2: Why are the error bars huge for K_M but small for k_{cat} ? How were the error bars actually computed?

Response:

The reviewer raises an important point. In this work, the error bars were computed by using a **bootstrap-based estimate**. The comparatively large error bars for K_M , in contrast to the small uncertainties for k_{cat} , arise from the different ways these parameters are constrained by the available data. In the Michaelis–Menten model, $v_0 = k_{cat}[E][S]/(K_M + [S])$, the value of K_M is determined primarily by the slope of the initial-rate curve at low substrate concentrations ($[S] \ll K_M$), where $v_0 \approx (k_{cat}[E]/K_M)[S]$. Thus, accurate estimation of K_M requires precise sampling in the low- $[S]$ regime. In our work, the simulations were conducted across a range of ATP concentrations from $\sim 10 \mu\text{M}$ to $\sim 1500 \mu\text{M}$, with only a subset of these concentration points fall within this critical low- $[S]$ region (e.g., $\leq 50 \mu\text{M}$). When applying the bootstrap procedure, each resampling trial selects a random subset of the available ($[S], v_0$) data points. Because K_M is highly sensitive to the inclusion or exclusion of individual low- $[S]$ points, the resulting distribution of bootstrap-fitted K_M values is broad, leading to larger error bars.

By contrast, k_{cat} is primarily determined from the plateau region of the Michaelis–Menten curve ($[S] \gg K_M$), where $v_0 \rightarrow k_{\text{cat}}[E]$. We sampled multiple high- $[S]$ conditions, and the saturated turnover rates across replicas exhibit relatively small statistical fluctuations. Consequently, the bootstrap-based estimate of k_{cat} is much more stable, resulting in smaller error bars.

- The authors write: "energy gap ΔV was optimized according to experimentally measured P_{closed} ." I suggest providing more details on how it was optimized.

Response:

We thank the reviewer's suggestions. The values of ΔV_i were optimized through an iterative procedure in which we systematically adjusted ΔV_i , performed molecular dynamics simulations, and computed the resulting population of the closed conformation. These simulated populations were then compared with the experimentally observed value ($P_{\text{closed}} = 0.3$). In the revised manuscript, we provide a more detailed description of this optimization procedure and the criteria used to determine the final ΔV_i values.

Corresponding changes to the manuscript:

(page 23, line 461) We added the following text in the revised manuscript to clarify our approach for determining the ΔV_i values:

"For the WT AdK, the ΔV_i values are optimized to reproduce the experimentally measured population of the closed state at apo condition ($P_{\text{closed}} = 0.3$)⁴⁴. The optimization procedure began with an initial guess of $\Delta V_i = 0$ Using this calibrated framework, we can systematically adjust the ΔV_i parameters to generate computational models of AdK variants with predetermined conformational preferences."

- Further, the authors state that "upon introducing crowder particles, the enzymatic activity monotonically diminishes with P_{closed} ." However, Fig. 3B shows a small peak at low P_{closed} . In Fig. 3A, not all curves seem to be in saturation, raising the question of how accurate the fitting was.

Response and Corresponding changes to the manuscript:

We thank the reviewer for pointing out this issue. Indeed, Fig. 3B shows a small peak in k_{cat} at low P_{closed} ; When P_{closed} is very small, conformational closing and productive substrate binding become rate-limiting steps. Crowding agents facilitate the conformational

closing process and thus may accelerate the catalytic cycle. However, within the range of parameters simulated, the increase in k_{cat} is minimal—almost within the margin of error of the model—so this trend is not emphasized here. Figure 4 illustrates this trend more clearly and provides a discussion of the underlying phenomenon. In Fig. 3A, the curves at low P_{closed} under dilute conditions are slightly unsaturated, which contributes to the slightly larger error bars observed in Fig. 3B. Nevertheless, despite these larger error bars, the difference in k_{cat} between the crowded and dilute conditions remains significant, and the overall conclusion regarding the effect of crowding on enzymatic activity remains robust.

- Fig. 4 shows that for $P_{\text{closed}} = 0.31$ and 0.54 , there is a non-monotonic behavior versus occupied volume fraction. I find this quite interesting and worth discussing in more detail in the manuscript.

Response:

We thank the reviewer for highlighting this point. As also noted by Reviewer 2, Fig. 4 in the original manuscript contained an error in which the axes were inadvertently inverted. This has been corrected in the revised manuscript. After correction, the non-monotonic dependence on crowding volume fraction is observed for the strongly open-biased AdK variants with low P_{closed} , rather than for the variants with $P_{\text{closed}} = 0.31$ and 0.54 .

This non-monotonic behavior arises from a shift in the rate-limiting step under crowding. For open-biased AdK, substrate binding is rate-limiting under dilute conditions. Moderate crowding stabilizes the closed conformation and accelerates substrate binding, leading to enhanced catalytic activity. However, at higher crowding levels, excessive stabilization of the closed state shifts the rate-limiting step from substrate binding to product release. The resulting slowdown in product release suppresses turnover, giving rise to the observed non-monotonic dependence of activity on crowding.

Corresponding changes to the manuscript:

We clarified this mechanism in the revised manuscript and expanded the discussion about Fig. 4 to explicitly connect the non-monotonic behavior to the crowding-induced shift in the rate-limiting step:

(page 12, line 247) We added following content in the revised the manuscript. “The effect of crowding depends strongly on the intrinsic conformational bias of the enzyme. For variants with $P_{\text{closed}} > 0.05$, increasing crowding leads to a monotonic reduction in enzymatic activity... this non-monotonic behavior arises because substrate binding must

precede catalysis. Moderate crowding stabilizes the closed conformation, thereby facilitating substrate binding. However, excessive crowding over-stabilizes closed conformations and slows product release, hindering completion of the catalytic cycle”

- The authors write that crowding "does not affect the relative population of the two pathways," but it is not substantiated, and it is difficult to judge based solely on Fig. 5A.

Response and corresponding changes:

We thank the reviewer for the comment. For accuracy, we have removed the statement that “crowding does not affect the relative population of the two pathways.”

- The authors state, "The acceleration of the substrate binding step primarily arises from the crowding enhanced open-to-closed conformational transition." I am unsure if I understand this. Crowding, quite generally, stabilizes the closed conformation. What does it mean that it enhances a transition? Quite generally too, crowding also enhances molecular binding (see e.g. <https://doi.org/10.1242/jcs.03063>), so perhaps this is the reason for the accelerated binding step?

Response and corresponding changes:

We thank the reviewer for this important question and for pointing out the need to clarify what we mean by “crowding-enhanced open-to-closed transition.” In the revised manuscript, we improved the description to clarify how crowding affects not only the relative stability of conformational states but also the height of the transition barriers between them.

As described in the manuscript, crowding reshapes the conformational energy landscape of AdK. It deepens the free energy well associated with the closed state, but it also modifies the barriers governing transitions between the open and closed conformations. Specifically, crowding lowers the barrier for the open-to-closed transition while increasing the barrier in the reverse direction. This asymmetric change results in a higher net transition rate toward the closed, binding-competent conformation. This is what we refer to as “enhancing the transition.”

Regarding the reviewer’s second point: in our implicit-ligand model, ligand binding is treated as a diffusion-limited Monte Carlo process governed by a fixed parameter k_{on} . Because we did not modify k_{on} when introducing crowding, our simulations do not explicitly account for crowding-induced changes to ligand diffusion rates. Therefore, the accelerated

substrate-binding step in our results does not arise from enhanced molecular diffusion or nonspecific crowding-induced affinity effects. Rather, it originates from the accelerated open-to-closed transition. Given that ligand binding in AdK is tightly coupled to domain closure, this shift in conformational dynamics directly speeds up the effective binding step in our model.

- In Fig. 6, there is no A, B, etc. Also, the figure caption is unclear (e.g., "starting from the open state for the product release")

Response and Corresponding changes:

We thank the reviewer for pointing out these issues. We have corrected the figure labeling in Fig. 6 and now include the appropriate panel identifiers (A, B, etc.). We have also revised the figure caption to make the descriptions unambiguous. In particular, the caption now clearly states that the trajectories shown in the product-release analysis were initiated from the closed conformation of holo-form AdK. These modifications improve both clarity and consistency in the presentation.

- The authors conclude that "adding crowding leads to opposite effects on the overall catalytic kinetics for the open-biased AdK model and closed-biased AdK model." As I already mentioned, Fig. 6 indicates a non-monotonic behavior, which I find even more interesting and suggest discussing. In the regime where crowding reduces the activity, Skora et al provided an analytical equation for the catalytic activity as a function of crowding (<https://doi.org/10.1039/D0CP06631A>). It would be interesting to see if this is consistent/fits to the authors' results.

Response:

We thank the reviewer for this insightful comment. In the theoretical study by Skora et al. (<https://doi.org/10.1039/D0CP06631A>), molecular crowding was also predicted to stabilize compact, closed conformations of enzymes, thereby hindering substrate binding. They reported that for physiologically relevant occupied volume fractions of 20–30%, enzymatic activity can be reduced by as much as 40–50%.

These predictions are in good agreement with our results. As shown in Figs. 2 and 4, for a broad range of AdK variants with $P_{\text{closed}} > 0.05$, physiologically relevant crowding conditions similarly lead to a reduction in enzymatic activity of approximately 50%. Although Skora et al. employed a highly simplified coarse-grained model, the consistency between their

theoretical predictions and our simulation results strengthens our conclusions. We have added a brief comparison to this work in the revised manuscript.

Corresponding changes to manuscript:

(page 13, line 259) We added the following text in the revised manuscript: “This result is consistent with the predictions of Skora *et al.*, who reported that at physiologically relevant occupied volume fractions of 20-30%, enzymatic activity can be reduced by 40-50%.”

- The authors should explain the meaning of a vertical line in $(r|r_0)$ in Eq. 2. They should also more clearly explain the model in this equation. For instance, how V_{exc}^{n-loc} is non-local if it depends solely on positions r ? The same concerns other terms.

Response:

The vertical line in the notation $(r|r_0)$ in Eq. 2 was intended merely as a visual separator between the current coordinates r and the reference (native) coordinates r_0 ; it carries no mathematical meaning (e.g., it does not denote a conditional probability). To avoid any potential confusion, we have replaced the vertical bar with a comma in the revised manuscript, so the expression now reads $V(r, r_0)$.

We have also clarified the physical meaning of the variables and the individual energy terms in Eq. 2. The coordinate vector \mathbf{r} represents the instantaneous positions of all coarse-grained particles in the model. Each term in the potential energy function (e.g., V_{bond} , V_{angle} , $V_{dihedral}$, etc.) depends on specific geometric quantities derived from \mathbf{r} , such as bond lengths, bond angles, dihedral angles, or inter-particle distances.

Regarding the term V_{exc}^{n-loc} , “non-local” refers to non-locality along the protein sequence. Specifically, this term acts only on residue pairs that are separated by more than a prescribed number of residues along the sequence and are not in close contact in the reference structure, thereby excluding bonded and near-neighbor, and native-contact interactions. For these non-neighboring residue pairs, a purely repulsive excluded-volume interaction of the form $(\sigma/r_{ij})^{12}$ is applied to prevent steric overlap and unphysical collapse of the chain.

Corresponding changes to the manuscript:

(page 21, line 430) We revised the explanation of Eq.2 in the manuscript. It now reads: “Here, r denotes the coordinates of the coarse-grained residues in a given structure, and

r0 refers to the reference coordinates from the native structure...Together, these terms produce a smooth, minimally frustrated energy landscape with a global minimum at the native state.”

- The authors further write that the apostate is decomposed into "3 double-basin systems describing the conformational motions between LID-CORE domains, NMP-CORE domains and LID-NMP domains." What does it mean: "a motion between the domains"? Motion of what? This must be better explained. Also, is this potential actually additive? I would suppose conformational changes in one domain can affect the changes in the other domain.

Response:

We thank the reviewer for pointing out this lack of clarity. By “conformational motion between domains,” we specifically refer to the relative movement of one domain with respect to another. In AdK, these motions correspond to changes in inter-domain orientations and distances, such as the opening and closing of the LID and NMP domains relative to the CORE domain, as well as their relative rearrangements with respect to each other. Accordingly, we introduced two separate double-basin energy terms to describe the LID–CORE and NMP–CORE motions, allowing the two substrate-binding domains to move independently. In addition, a third double-basin energy term was added to describe interactions at the LID–NMP interface, capturing the coupling between the motions of the two domains. The total potential energy of the protein is therefore given by the sum of these three components. The movement of shared residues among these double-basin systems couple the motions of different domains.

We have revised the manuscript to clarify the definition of “motion between domains,” the rationale for the decomposition into multiple double-basin subsystems, and the physical interpretation of the additive potential.

Corresponding changes to the manuscript:

(page 22, line 454) We revised the explanation of double-basin energy function construction in the manuscript. It now reads: “The main conformational changes in AdK involve opening and closing motions of the LID and NMP domains relative to the CORE domain...The movement of shared residues among these double-basin systems couple the motions of different domains.”

- Concerning Eq. (3), do I understand correctly that it means adding a binding energy between residuals of the enzyme that would be in contact with a ligand if the ligand were explicitly modelled? This piece must be clarified.

Response and Corresponding changes:

We thank the reviewer for this helpful advice. The reviewer is right, upon a binding event, Eq. (4) accounts for interactions involving residues at the binding site that would contact the ligand if it were explicitly modeled. In the revised manuscript, we have clarified this point.

(page 23, line 477) The following text was added: “The energetic contribution of ligand binding at site l_i is described by the potential: ... where the summation spans residue pairs (I, J) that would normally interact with the ligand if the ligand were explicitly included at binding site l_i .”

- The simulation method (as I already mentioned) must be more clearly explained. For instance, regarding Eq. (4), do I understand correctly that with the probability following from Eq. (4), some residuals of the enzyme that would face a ligand (if it were explicitly modeled) become bound with the potential given by Eq. (3)? Why this particular form of Eq. (4)? Similarly, what motivates the choice of the expression for k_{off} given by Eq. (5)? One does not have to provide derivations but giving some clear physical reasoning would be very helpful.

Response and Corresponding changes:

We thank the reviewer for this insightful comment. In the revised manuscript, we have significantly expanded the description of our implicit ligand modeling scheme to clarify the physical rationale behind Eqs. (4) and (5).

To directly answer the reviewer’s question: yes, when a binding event is accepted according to Eq. (4), the residues at the binding site that would interact the ligand if it were explicitly modeled. Since the ligand is not represented by physical particles in our approach, the protein-ligand interaction is instead modeled by V_{bind} (Eq. 3), which imposes ligand-mediated residue–residue interactions in the binding site based on their structural proximity.

Eq. 4 defines the acceptance probability for binding event, derived from a kinetic Monte Carlo framework under the assumption of diffusion-limited association. Specifically, the on-

rate is modeled as $k_{on} = k_D[L]$, where k_D is the diffusion-controlled second-order rate constant and $[L]$ is the effective ligand concentration. This form avoids the need to simulate explicit ligand diffusion. Conversely, Eq. 5 governs the unbinding probability and is motivated by transition state theory: the off-rate scales with the Boltzmann factor of the current binding energy, $k_{off} \propto \exp[-V_{bind}/k_B T]$. This captures the intuitive idea that deeper energetic stabilization makes dissociation less probable.

Together, these choices allow us to model ligand binding/unbinding as stochastic transitions between conformational states, driven by both ligand concentration and instantaneous protein–ligand compatibility—without introducing explicit ligands or sacrificing computational efficiency. We now explain this logic explicitly in the Methods section of the revised manuscript.

(page 24, line 488) We have expanded the explanation of the implicit-ligand model in the revised manuscript.

- If I am not mistaken, upon successful MC steps describing ligand binding, the interaction potentials are dynamically changed. Doesn't it generate numerical instabilities in MD simulations due to large forces (potentially emerging because the system is not "equilibrated" for this potential)?

Response:

We thank the reviewer for raising this important concern. In our model, the introduction of ligand-binding interactions upon successful MC binding events does not lead to numerical instabilities. This is because the ligand-mediated interactions are described by smooth, Gaussian-shaped attractive potentials without an explicit repulsive component (Eq. 4 in the main text), which avoids the generation of large forces. As a result, the system remains numerically stable. This approach has been used extensively in our previous simulations, where no stability issues were observed.

- The authors impose a box potential on their system. Why not use periodic boundary conditions? Does this box potential affect the enzyme kinetics?

Response:

We thank the reviewer for raising this important point. In our simulations, we used a soft confining (box) potential rather than PBC to mimic a locally crowded and confined

environment around the enzyme, whereas PBC conditions represent an effectively infinite system.

Regarding the potential impact on enzyme kinetics: in all our simulations, AdK was initially placed near the center of the simulation box and surrounded by crowding agents. Due to the viscous, crowded environment—which strongly suppresses long-range diffusion—the enzyme rarely approached the boundaries of the box. As a result, the confining potential had negligible influence on its conformational dynamics or binding kinetics.

- Providing some more details on MD simulations would also be helpful, e.g., on the CafeMol time units. I am somewhat confused how the authors managed to reach 112 milliseconds. With how many beads was the enzyme represented?

Response:

Calibrating the time step in CG simulations is inherently challenging. Because the CG model reduces the degrees of freedom and simplifies the underlying energy functions, the simulation time unit cannot be directly matched to real physical time. In our framework, we mapped the CG timescale to the experimental one by comparing the rates of conformational motions of apo AdK obtained from CG simulations with the experimentally measured rates. Based on this calibration, the total simulation length of 2×10^8 MD steps corresponds to approximately 112 ms in real time. Regarding the coarse-grained representation, each amino acid residue is modeled as a single bead, resulting in 214 beads for AdK. We have added these clarifications to the Methods section for improved transparency.

Corresponding changes to the manuscript:

(page 27, line 555) The following sentences were added in the main text of this revised manuscript: “Given the fact that our CG model reduces molecular complexity by simplifying degrees of freedom and energy functions, the time unit (τ) does not directly correspond to physical time. To establish a meaningful kinetic timescale, we calibrated the simulation dynamics against experimental data by matching the rate of conformational transitions in the apo state of AdK in our previous works. Based on this calibration, 2×10^8 MD steps correspond to approximately 112 milliseconds of real time”

- In Supplementary materials, R in Eq.(1) is not defined. Also, where does the form of Eq.

(1) follow from? Similarly, I suggest providing more information on the choice of the reaction coordinate, Eq. (3), in the SM.

Response:

We thank the reviewer for pointing out this unclear description. In Eq. 1, \mathbf{R} should indeed be \mathbf{r} , representing the Cartesian coordinates of the coarse-grained beads.

The construction of the double-basin potential follows the approach developed by Okazaki et al [PNAS, 2006,103 (32) 11844-11849]. In this framework, the double-basin potential V_{DB} is obtained by analogy with the quantum mechanical treatment of electron transfer. Specifically, a smooth two-state potential is defined as the lower eigenvalue of the

characteristic equation:
$$\begin{pmatrix} V(\mathbf{r}, \mathbf{r}_1) & \Delta \\ \Delta & V(\mathbf{r}, \mathbf{r}_2) + \Delta V \end{pmatrix} \begin{pmatrix} c_1 \\ c_2 \end{pmatrix} = V_{DB} \begin{pmatrix} c_1 \\ c_2 \end{pmatrix}$$

$V(\mathbf{r}, \mathbf{r}_1)$ and $V(\mathbf{r}, \mathbf{r}_2)$ are the single-basin AICG2+ energy functions corresponding to the two reference structures—open and closed states in our study. The variable \mathbf{r} denotes the Cartesian coordinates of the CG beads, while \mathbf{r}_1 and \mathbf{r}_2 are the coordinates of the two reference states. The parameter Δ is the coupling constant, and (c_1, c_2) is the eigenvector associated with the eigenvalue V_{DB} .

A nontrivial solution exists only when the secular equation:

$$\begin{vmatrix} V(\mathbf{r}, \mathbf{r}_1) - V_{DB} & \Delta \\ \Delta & V(\mathbf{r}, \mathbf{r}_2) + \Delta V - V_{DB} \end{vmatrix} = 0$$

is satisfied. The lower-energy eigenvalue is used as the double-basin potential V_{DB} . Because this potential is continuous and differentiable, it can be directly employed in MD simulations. The associated eigenvector (c_1, c_2) also provides a convenient reaction coordinate for monitoring state transitions; we use $\chi = \ln(c_2/c_1)$ to indicate whether the system resides closer to basin 1 or basin 2. This interpolation scheme is particularly useful because it allows independent tuning of two fundamental energy scales: the coupling constant Δ directly modulates the height of the transition barrier, whereas ΔV sets the relative stability between the two basins.

We have added additional explanation in the revised manuscript to improve clarity.

- The authors write: "Three double-basin energy terms are employed to describe the interaction of each pair of two domains of AdK." If there are two domains, there is only a single pair.

Response:

We thank the reviewer for the comments. AdK's LID and NMP domains undergo opening and closing motions relative to the CORE domain. To capture these motions, we partitioned the protein into three separate double-basin energy terms: one for LID–CORE motions, one for NMP–CORE motions, and one for interactions at the LID–NMP interface. This scheme allows the two substrate-binding domains, LID and NMP, to move independently while also accounting for the coupling between them.

Corresponding changes to the manuscript:

(page 22, line 454) In the revised manuscript, we changed the expression: "The main conformational changes in AdK involve opening and closing motions of the LID and NMP domains relative to the CORE domain...The movement of shared residues among these double-basin systems couple the motions of different domains."

- This sentence in the SM is not fully clear: "When a substrate or product was thought to bind to AdK, bunches of ligand-mediated contacts around the ligand binding site were added to the AICG2+ energy function to stabilize the closed conformation."

Responses:

We thank the reviewer for pointing out this lack of clarity. As suggested in the reviewer's previous comments, we have expanded and clarified the description of the implicit-ligand model in the revised main text to more clearly explain how ligand-mediated contacts are introduced to stabilize the closed conformation.

- Eq. (4) in the SM, V is the extra energy contributed by the ligand-mediated contacts. What is this extra energy? Is it a function? Is it a value? Why in Eq. (3) and (4) in the SM, there is a proportionality sign, while the same equations in the main text have equality. Seeing the difference, it would nevertheless be good to keep consistency between the SM and the main text.

Response and Corresponding changes to the manuscript:

We thank the reviewer for highlighting these important points regarding our implicit ligand model. Regarding Eq. (4) in the SM: the term V represents the total energetic contribution from all ligand-mediated contacts. It is a function—specifically, the sum of Gaussian-based attractive potentials for each added contact—and is added to the baseline AICG2+

Hamiltonian. We acknowledge the inconsistency in equation expression between the main text and the SM. To avoid confusion, we have now standardized the notation across both documents. These changes have been implemented in the revised Supporting Information.

- On page 3 of the SM, the authors refer to figure S1, but it does not seem to be the correct figure.

Response and Corresponding changes to the manuscript:

We thank the reviewer for catching this error. We have corrected the figure reference in the revised Supporting Information.

- Various parameters in Eq. (5) of the SM are also not defined.

Response and Corresponding changes to the manuscript:

We appreciate the reviewer's careful reading. In the revised manuscript, we have added clear definitions for all parameters appearing in Eq. (5) to improve transparency and readability

Reviewer #2 (Remarks to the Author):

This manuscript addresses the timely and complex question of how molecular crowding influences enzymatic catalysis, a topic of broad significance in biophysics. The authors employ adenylate kinase (AdK), a model system extensively studied in both experimental and computational contexts, to explore how inert crowders reshape the conformational equilibrium and impact individual steps of the enzymatic cycle. Using coarse-grained simulations, they provide quantitative insights into how crowding stabilizes particular conformational states and thereby modulates turnover, substrate binding, and product release. Importantly, the study reveals that crowding can either enhance or inhibit activity depending on the rate-limiting step of the cycle, offering a mechanistic framework that reconciles diverse prior observations. Overall, this is a carefully executed and conceptually significant work that will be of interest to a broad readership. However, there are several important issues—mainly concerning the clarity and presentation of the manuscript—that the authors should address before it can be considered for publication. I expect that

resolving these points should be relatively straightforward.

Response:

We appreciate the reviewer's positive feedback of our work

Major comments

1. There appears to be some confusion regarding the definition of $P(\text{closed})$. Initially, it is defined as the relative population of the closed conformation. Later in the manuscript, in particular in Figure 4, this definition seems inverted, with high values corresponding to an open-biased conformation. In addition, the trends in Figures 3 and 4 appear to be opposite: while in Figure 3 adding crowder at low $P(\text{closed})$ enhances activity, in Figure 4 the reverse seems to be the case. A low $P(\text{closed})$ is also designated as closed-biased in the cartoon, suggesting that the panels in Figure 4 may have been accidentally swapped. Similarly, in line 206 conformations with $P > 0.3$ are described as open-biased, where closed-biased would be correct. Clarifying the definition of $P(\text{closed})$ and checking the consistency of figures and text would greatly improve readability.

Response and Corresponding changes to the manuscript:

We thank the reviewer for carefully identifying this important inconsistency. The reviewer is correct that, due to an oversight, the labeling of P_{closed} in Fig. 4 was inadvertently swapped, leading to confusion in both the figure and the associated text. We have corrected Fig. 4 and thoroughly checked the manuscript to ensure that the definition and usage of P_{closed} are now consistent throughout. The revised manuscript reflects these corrections.

2. The authors investigated not only inert crowders but also attractive interactions. However, these results are not discussed in the RESULTS section and can only be found in the SI, despite being referred to multiple times in the manuscript and discussed in detail in the DISCUSSION. These data could provide an interesting link to experimental findings mentioned by the authors, such as the activity enhancement by urea, which is notable given urea's typical role as a denaturant. I recommend that the authors include at least a brief summary of the main results for attractive crowders in the RESULTS section.

Response and Corresponding changes to the manuscript:

We appreciate the reviewer's constructive suggestion. In the revised manuscript, we added

a new subsection in the Results section to briefly discuss the role of crowder size and attractive (soft) crowder-enzyme interaction in regulating AdK's catalytic dynamics.

3. Figure 6 lacks panel labels (A, B, etc.), which makes it difficult to follow the discussion in the text. Furthermore, the authors state: "The acceleration of the substrate binding step primarily arises from the crowding enhanced open-to-closed conformational transition" (lines 252–253), but the figure does not actually show the conformational transitions. Also, it would also be helpful to refer to Figure 6 earlier in the text, no later than line 250.

Response and Corresponding changes to the manuscript:

We thank the reviewer for raising this important point. In the revised manuscript, Fig 6 is improved. As explained in the revised main text and Supporting Information, in our implicit-ligand model the ligand-binding rate depends on ligand concentration and ligand diffusion, which is assumed to be constant in the present study. In contrast, the ligand-release rate is primarily governed by the binding energy. Consequently, the observed changes in substrate binding and release rates under crowding conditions mainly arise from crowding-induced modulation of domain opening and closure dynamics.

As suggested by the reviewer, we now refer to Fig. 6 earlier in the revised manuscript.

4. The authors consider $P(\text{closed})$ as a critical determinant of activity. However, for equilibrium populations to be the controlling factor, the conformational transitions themselves must be fast and not rate-limiting. The authors should clarify this prerequisite in the text.

Response and Corresponding changes to the manuscript:

We thank the reviewer for raising this important point.

(page 7, line 136) In the revised manuscript, we added the following statement to explicitly clarify this assumption: "In addition, the rate-limiting step in the AdK catalytic cycle is ligand unbinding coupled with conformational transition."

5. The reported changes in K_M are small (60 to 70 μM) and fall within the error bars (lines 165–171). I would therefore consider K_M to be unaffected by crowding, in contrast to the clear changes observed in k_{cat} . From a theoretical standpoint, the only mechanism by which K_M might change is if the various substrate-bound species are affected differently by crowding (see next point), thus manipulating substrate binding and dissociation.

Response and Corresponding changes:

We thank the reviewer for pointing this out. As suggested in the following comment, we computed the formation probability of the productive substrate-bound state of WT AdK under various crowding conditions (Fig. R2). Moderate crowding slightly promotes substrate binding, whereas high crowding suppresses productive substrate binding by favoring non-productive ligand-bound states. The trend is consistent with the change of K_M with crowding. We agree that the observed changes in K_M are small and fall within the fitting uncertainties. Therefore, we state that the interpretation of the apparent differences in K_M should be treated with caution.

6. The conformational equilibrium of AdK is known to be strongly influenced by substrate binding. The authors mention nine different substrate-bound species, but do not discuss whether crowding has distinct effects on these species. Among other places, this could be relevant for the discussion in lines 171–175: while crowding may hinder substrate binding by favoring the closed state in the apo form, it could also strengthen ligand binding if primarily the substrate-bound species are stabilized.

Response and Corresponding changes to manuscript:

We thank the reviewer for this valuable suggestion. Following the reviewer's recommendation, we calculated the formation probabilities of the catalytically competent state under various crowding conditions at [ATP]=9.5 μ M. Moderate crowding slightly stabilizes the closed conformation, which exhibits stronger substrate binding, consistent with a small decrease in K_M . As the reviewer expected, high crowding over-stabilizes the closed conformation and begins to suppress productive substrate binding while promoting nonproductive binding, consistent with an increase in K_M . However, the observed changes in K_M are small relative to their large uncertainty; therefore, this trend should be interpreted with caution. We have incorporated this analysis and discussion in the revised manuscript (lines 194-203).

Figure R2. Formation probabilities of the catalytically competent state (substrate-bound closed state) of WT AdK at [ATP] = 9.5 μ M under various crowding conditions. The low probabilities primarily reflect the rapid conversion of the catalytically competent state during the reaction cycle.

Minor comments

1. Line 22: Please provide a reference for the specific number (30%).

Response and corresponding changes:

We have added references describing the typical volume fraction of crowded cellular environments.

2. Figure legend 1: In panel 1B, were the substrates modeled into the crystal structure of the protein? Otherwise, the shown molecule would be AP5. Second, the sentence "AMP can nonspecifically bind at the AMP site but with lower affinity" is unclear; I assume the ATP site is meant. Experimental evidence argues against competition between ATP and AMP at the ATP site. Since AMP binding at the ATP site is not considered later in the paper, I recommend removing this sentence.

Response and corresponding changes:

We have followed the reviewer's suggestion by modifying Figure 1B to display the AP5 ligand in the crystal structure rather than showing ATP and AMP as in the original figure. Additionally, we have removed the unclear statement regarding AMP binding ("AMP can nonspecifically bind at the AMP site but with lower affinity") from the revised manuscript, as suggested by the reviewer."

3. Line 183: Please provide a reference for the value of 0.3.

Response and corresponding changes:

We added the references for the value of 0.3

4. Line 196–197: “In dilute solution, the k_{cat} value exhibits a biphasic trend. It increases when $P_{closed} < 0.05$ and decreases when $P_{closed} > 0.05$.” It would be clearer to state that k_{cat} is maximal at $P_{closed} \sim 0.5$.

Response and corresponding changes:

Following reviewer’s suggestion, we explicitly express that k_{cat} is maximal at $p_{closed} \sim 0.05$ in the revised manuscript.

5. Line 205: “decreases” would be correct instead of “increases.”

Response and corresponding changes:

We thank for the reviewer’s comment. We correct the mistake in the manuscript.

6. Figure legend 3: I assume that all lines, not only the solid ones, correspond to MM fits? Please clarify.

Response and corresponding changes:

We have revised the caption of Figure 3, and state that all lines are MM fits.

7. Line 233: It would be helpful to mention the substrate concentrations in the text, not only in the figure legend.

Response and corresponding changes:

We have explicitly expressed the substrate concentrations in the text in the revised manuscript.

8. Lines 243–244: The statement “Notably, more substantial enhancement is observed for the AdK variants with moderate P_{closed} values (e.g., 0.3)” is expected, since the energy difference between the states is lowest at moderate P values.

Response and corresponding changes:

We thank the reviewer for the helpful suggestion. We removed the statement in the revised manuscript

9. Line 250: From Figure 6, product release appears to be “insensitive” rather than “less sensitive” to substrate concentration.

Response and corresponding changes:

We replaced the “less sensitive” with “insensitive” in the revised manuscript.

10. Lines 316–337: The phrase “Consistent with experimental observation” should be clarified. The experimental references cited in the Introduction mainly concern shifting the conformational equilibrium by mutations or other means, not by inert crowding. A specific reference should be provided, or the sentence adjusted.

Response and corresponding changes:

We thank the reviewer for this suggestion. For accuracy, we removed the statement “Consistent with experimental observations” from the revised manuscript.

11. Figure 7: In light of major comment 6, the authors might consider splitting the first step into substrate binding and conformational transition. Faster closing does not promote turnover in the apo protein, as it can hinder substrate binding, but once substrates are bound, faster closing accelerates the reaction.

Response and corresponding changes:

We thank the reviewer for this valuable comment. In the revised manuscript, we have updated Fig. 7 by explicitly separating the first step into substrate binding and the subsequent conformational transition. The revised figure illustrates that molecular crowding can accelerate domain closure of the substrate-bound enzyme starting from the open conformation.

12. SI page 3: The reference in the sentence “This classification was employed to characterize the kinetics of AdK’s functional dynamics, encompassing processes such as substrate binding, product release, and the catalytic cycle (see Fig S1)” is misleading, as Fig. S1 does not show this. Please correct the figure reference.

Response and corresponding changes:

We thank the reviewer for identifying this error. We have corrected the reference from “(see Fig S1)” to “(see Fig 1 in main text)” in the revised SI.

13. Overall, the message of the manuscript is clear and easy to follow. However, several

typographical and minor errors slightly impair readability. I recommend a careful proofreading to correct these.

Response:

We sincerely thank the reviewer for pointing out the errors, missing citations, and other oversights, as well as for providing valuable suggestions. Following these comments, we have made the corresponding corrections and revisions in the updated manuscript.

Responses and the corresponding changes

We gratefully acknowledge the reviewers' positive feedback on our revised manuscript. All comments, questions, and concerns have been carefully and comprehensively addressed. A detailed point-by-point response, together with the corresponding revisions, is provided below. The revised text is highlighted in blue in the manuscript.

Reviewer #1 (Remarks to the Author):

The authors have quite carefully addressed the referees' comments. I have only a few comments that I suggest addressing before publication.

Response: We sincerely thank the reviewer for the positive evaluation and constructive feedback on our revised manuscript. We have addressed the additional comments and provide our detailed responses below.

- The authors write that "a key advance" of their work is the integration of "the enzyme model with a macromolecular crowding model". This appears somewhat straightforward to me, however. In my opinion, the main advance is the investigation of crowding effects on complex enzymatic reactions, which are still rare in the literature.

Response: We thank the reviewer for this valuable insight. We have revised the manuscript to emphasize this aspect and clarify its significance.

Corresponding changes to the manuscript:

(page 5, line 113) We revised the sentence in the manuscript to read: "Building on this model, here we further construct a computational framework with explicit consideration of the crowding agent and use it to investigate the influence of crowding on complex enzymatic reactions, which have been largely overlooked in current studies."

- The authors write that "moderate crowding shifts the conformational equilibrium toward this closed state, which can enhance effective substrate affinity" (also in the main text: "Moderate crowding stabilises the closed conformation and accelerates substrate binding"). Why does stabilising closed configurations accelerate substrate binding? Or why would the intrinsic affinity increase? If the probability of the closed conformation increases, one could actually expect the opposite (as the authors write for strong crowding conditions). Perhaps closing stabilises the bound state (hindering dissociation), but only provided a substrate does bind to the active site, and through this stabilization enhances reactivity? For stronger crowding, the binding is of course hindered, per se. These points should be clarified.

Response: We thank the reviewer for pointing this out. We acknowledge that our original expression was imprecise. By 'accelerated substrate binding,' we specifically mean the faster formation of the catalytically competent state (which requires both substrate binding and domain closing). Under moderate crowding, the closed state is more stable and binds substrate more tightly, leading to higher effective affinity. Additionally, moderate crowding speeds up the closing motion itself, allowing the enzyme to reach this active state faster. However, as the reviewer correctly noted, under strong crowding, the enzyme closes too quickly often before the substrate binding, leading to reduced affinity and slowdown of the formation of the active complex.

Corresponding changes to the manuscript:

(page 13, line 262) we added the following to the revised manuscript: “Moderate crowding stabilizes the closed conformation, shifts the conformational equilibrium toward this catalytically competent state. Additionally, it speeds up the domain closing motion itself. These effects enhance the probability of forming the active, substrate-bound complex, thereby accelerating the overall rate of reaching the catalytically competent state (i.e., the apparent rate of substrate binding). In contrast, excessive crowding over-stabilizes the closed conformation. This causes the enzyme to close too quickly often before the substrate binding, and impedes product release. Consequently, these factors hinder effective substrate binding and disrupt the completion of the catalytic cycle.”

- In some cases, the authors provide the response, but it is not taken into account in the manuscript. For instance, regarding the error bars, the authors added in the captions that they are based on bootstrapping. However, perhaps I missed it, but I do not see any explanation in the manuscript for the large error bars for K_M . This might be useful to readers. Similarly, the impact of the confining box on enzyme kinetics, particularly the fact that the enzyme does not approach the box boundaries within typical simulation time scales, is also important.

Response: We thank the reviewer for pointing this out. In the revised manuscript, we addressed these points explicitly.

Corresponding changes to the manuscript:

1. (page 10, line 203) We added the following statement in the revised manuscript: “It is worth noting that the large uncertainty in K_M is primarily attributable to fitting errors. According to the Michaelis-Menten equation, K_M is predominantly determined by data points at low substrate concentrations. In our study, however, measurements at such low concentrations were limited. Consequently, the bootstrap analysis used to estimate uncertainty yields relatively large errors, leading to the observed variability.”

2. (page 27 line 568) We added the following statement in the revised manuscript: “It should be noted that we employed a soft confining (box) potential rather than periodic boundary conditions (PBC) to mimic a locally crowded and confined environment around the enzyme, whereas PBC effectively represents an infinite system. In all simulations, AdK was initially positioned near the center of the simulation box and surrounded by crowding agents. Due to the viscous, crowded environment, which strongly suppresses long-range diffusion, the enzyme rarely approached the box boundaries. Consequently, the confining potential exerted negligible influence on the enzyme’s conformational dynamics or binding kinetics.”

- Regarding three double-basin systems, I assume this representation is an approximation, so I suggest stating this explicitly in the manuscript.

Response: We thank the reviewer for this suggestion. We have clarified in the manuscript that the double-basin systems are an approximation of the functional energy landscape.

Corresponding changes to the manuscript:

(page 23, line 470) We revised the sentence in the manuscript to read: “This approach generates a multi-basin energy landscape that approximates the complex functional energy landscape of proteins and explicitly accounts for conformational interconversion”.

- The authors avoid explicit simulations of ligands. As I understand, this is a trick simplifying and speeding up the simulations. However, ligand diffusivity can also be affected by crowding, which would influence overall enzymatic kinetics. It seems this effect is not taken into account within the authors’ approach. Some studies indicate quite a significant slowdown of even small molecules (see <https://doi.org/10.1002/jmr.709>, <https://doi.org/10.1021/acs.nanolett.3c05100>,

<https://doi.org/10.7554/eLife.19274>). How would that effect impact the authors' results? I suggest commenting on this point in the manuscript.

Response: We thank the reviewer for highlighting the impact of crowding on ligand diffusion. While our current model assumes fast substrate binding relative to domain motion, and neglects the slowdown of substrate diffusion, we agree that severely reduced diffusivity could become rate-limiting and alter the overall kinetics.

Corresponding changes to the manuscript:

(page 20, line 393) We have added a discussion to acknowledge this point and suggest it as an important direction for future work. "On the other hand, several studies have demonstrated that macromolecular crowding can significantly slow down substrate diffusion, leading to the inhibition of enzymatic catalysis^{27,74-76}. This effect of crowding on substrate diffusion is neglected in our current model, in which substrate diffusion coefficient is assumed to be constant. However, we anticipate that if substrate diffusion were dramatically slowed to become comparable to or even slower than domain motion, the crowding effect on substrate diffusion would become a non-negligible factor influencing catalysis. In particular, if substrate diffusion becomes the rate-limiting step, it would dominate the overall reaction kinetics and potentially overshadow the conformational dynamics effects reported in this study. This represents an important consideration for future investigations extending our framework to more extreme crowding conditions."

Reviewer #2 (Remarks to the Author):

The authors have addressed all my concerns with great effort and care, and I recommend the publication of their work. I have two minor recommendations the authors might consider for the final manuscript.

Response: We sincerely appreciate the reviewer's positive evaluation and their recommendation for publication.

1) I would appreciate it if explicit numbers for the transition rates were given, for an easier comparison with the explicit numbers for turnover. Line 561 would be a good spot, in my opinion. Also including units in Table S1 would be better.

Response:

We thank the reviewer for this helpful suggestion. As recommended, we have added the explicit transition rate values for wild-type (WT) AdK in the apo state. This addition facilitates a direct comparison with the turnover numbers. Additionally, we have updated the caption of Table S1 to explicitly include the units for ΔV (kcal/mol), ensuring clarity for all reported parameters.

Corresponding changes to the manuscript:

(page 28, line 595) We have inserted the following sentence: "The transition rate for WT AdK in the apo state was determined to be $\sim 7.0 \text{ ms}^{-1}$."

2) The revised Fig. 7 is inaccurate, as it suggests that closing might occur before substrate binding. I recommend removing the arrow between the non-substrate-bound closed conformation and the bound closed conformation.

Response and corresponding changes to the manuscript:

We thank the reviewer for pointing out this inaccuracy. We agree that the original arrow could misleadingly suggest that domain closing occurs prior to substrate binding. To address this, we have

removed the arrow connecting the non-substrate-bound closed conformation and the substrate-bound closed conformation in Fig. 7